# CONVERGENCE ANALYSIS AND IMPLICIT REGULARIZATION OF FEEDBACK ALIGNMENT FOR DEEP LINEAR NETWORKS

## ABSTRACT

We theoretically analyze the Feedback Alignment (FA) algorithm, an efficient alternative to backpropagation for training neural networks. We provide convergence guarantees with rates for deep linear networks for both continuous and discrete dynamics. Additionally, we study incremental learning phenomena for shallow linear networks. Interestingly, certain specific initializations imply that negligible components are learned before the principal ones, thus potentially negatively affecting the effectiveness of such a learning algorithm; a phenomenon we classify as implicit anti-regularization. We also provide initialization schemes where the components of the problem are approximately learned by decreasing order of importance, thus providing a form of implicit regularization.

## 1 INTRODUCTION

In order to train a Machine Learning (ML) architecture in the context of supervised learning, it is a consolidated practice to resort to the Gradient Descent (GD) algorithm and its variants (e.g. stochastic GD) and to calculate the gradient of the loss function via backpropagation (Rumelhart et al., 1986). However, one of its limitations is the so-called weight transport problem (Grossberg, 1987): the update of each neuron during the learning phase relies on the downstream weights (which are therefore "transported" along the backward pass). This implies that 1) the synaptic weights are the same, up to transposition, for both the forward (inference) and backward (learning) pass and 2) each neuron has a precise knowledge of all of its forward synapses. It is known that such a model does not accurately reflect the behaviour of a human brain, thus it is biologically implausible (Crick, 1989). Feedback Alignment (FA), originally proposed by Lillicrap (Lillicrap et al., 2016), has been an attempt to mitigate this discrepancy between the neuroscience side and the engineering side of AI and to tackle the weight transport problem in a systematic way. The simple, yet powerful idea is to replace the transposed weight matrices in the backpropagation algorithm with arbitrary, fixed matrices: via this strategy, each neuron during training receives a random projection of the error vector.

In order to illustrate the algorithm and for the purpose of our study, let us consider a fully-connected $L$-layer neural network (NN): given the input, $\boldsymbol{x} = \boldsymbol{h}_0 \in \mathbb{R}^d$, the predicted label $\hat{\boldsymbol{y}} \in \mathbb{R}^o$ is computed as

$$\boldsymbol{a}_\ell = \boldsymbol{W}_\ell \boldsymbol{h}_{\ell-1}, \quad \boldsymbol{h}_\ell = \sigma(\boldsymbol{a}_\ell), \quad \ell \in \{1, \dots, L\} := [L], \tag{1}$$

$\sigma$ being a (potentially) nonlinear function applied entry-wise to the vector $\boldsymbol{a}_\ell$, and finally $\hat{\boldsymbol{y}} = f(\boldsymbol{a}_L)$ for some output nonlinear function $f$. We consider a regression task with loss function $\mathcal{L}$ being the standard Mean Square Error (MSE). The backpropagation strategy for GD prescribes the following updates for each layer in the network: given a step size $\eta > 0$ and the error vector $\boldsymbol{e} = \delta\boldsymbol{a}_L = \frac{\partial \mathcal{L}}{\partial \hat{\boldsymbol{y}}} = \hat{\boldsymbol{y}} - \boldsymbol{y}$ where $\boldsymbol{y}$ is the true label, we have

$$\delta\boldsymbol{W}_L = -\eta \boldsymbol{e}\boldsymbol{h}_{L-1}^\top, \quad \delta\boldsymbol{W}_\ell = -\eta\delta\boldsymbol{a}_\ell \boldsymbol{h}_{\ell-1}^\top \quad \text{and} \quad \delta\boldsymbol{a}_\ell = \left(\boldsymbol{W}_{\ell+1}^\top \delta\boldsymbol{a}_{\ell+1}\right) \odot \sigma'(\boldsymbol{a}_\ell), \ \ell \in [L-1] \tag{GD}$$

where $\odot$ denotes the Hadamard product. The FA recipe prescribes the following:

**Feedback Alignment:**   in (GD) use   $\delta\boldsymbol{a}_\ell = (\boldsymbol{M}_\ell \delta\boldsymbol{a}_{\ell+1}) \odot \sigma'(\boldsymbol{a}_\ell), \quad \ell \in [L-1].$   (2)

where $\{\boldsymbol{M}_\ell\}$ are a collection of arbitrarily chosen matrices, which do not evolve during training.

**Related and previous work.** Inspired by Lillicrap et al. (2016) and shortly after its publication, Nøkland (2016) proposed two new algorithms called Direct and Inverse FA. In particular, Direct FA reduces to the FA algorithm in the case of a 2-layer NN or multi-layer linear NN; the difference between these two algorithms in the general multi-layer nonlinear NN resides in injecting the error vector directly into the update of each hidden layer of the network: $\delta a_\ell = (M_\ell e) \odot \sigma'(a_\ell)$, $\ell \in [1, L-1]$. Such a modification of the FA algorithm can be interpreted as a noisy version of layer-wise training (Gilmer et al., 2017).

Since its first formulation, there has been an extensive numerical analysis on testing whether FA and Direct FA algorithms can be successfully applied to ML problems, in particular in the presence of complex datasets and deep architectures (Bartunov et al., 2018; Moskovitz et al., 2018; Launay et al., 2019). We mention the recent survey paper Launay et al. (2020) where the focus is on Direct FA and it is shown that the algorithm performs well in modern deep architectures (Transformers and Graph NN, among others) applied to problems like neural view synthesis or natural language processing. Furthermore, more on the applied (biologically-inspired) side, there have been recent pushes in furthering the idea of FA: spike-train level direct feedback alignment (Lee et al., 2020), memory-efficient direct feedback alignment (Chu et al., 2020) and direct random target projection (Frenkel et al., 2021) to mention a few examples.

On the other hand, rigorous theoretical results are scarce. Preliminary asymptotic results can be found in the original papers by Lillicrap et al. (2016) and Nøkland (2016). More recently, Refinetti et al. (2020) presented the derivation of a system of differential equations that governs the Direct FA dynamics for a 2-layer non-linear NN in the regime where the input dimension goes to infinity. There is no explicit formula for the solution of the system, but via a careful local analysis it is possible to identify two separate phases of learning (alignment and memorization) and a "degeneracy breaking" effect: at the end of the FA training, the selected solution maximizes the overlap between the second weight matrix $W_2$ and the FA matrix $M$. Such an "alignment" dynamics is also evident in our analysis (Section 2). Convergence guarantees for deep networks are still missing from the literature and one of the goals of this paper is indeed to provide rigorous proofs of convergence, together with convergence rates.

Our analysis is in a similar vein as some recent works on implicit regularization in linear neural networks Saxe et al. (2018); Gidel et al. (2019); Arora et al. (2018). Even though our setting is very similar, our analysis is different: because of the feedback alignment matrices, the autonomous differential equation obtained significantly differ and lead to different dynamics. Moreover our focus is slightly different from Saxe et al. (2018); Gidel et al. (2019); Arora et al. (2018) which are fully dedicated to implicit regularization of GD while we also aim at proving the convergence of FA.

**Our contributions.** We aim to provide a theoretical understanding of how and when FA works. We focus on two aspects: 1) proving convergence of the algorithm and 2) analyzing implicit regularization phenomena. We will extensively study the case of *deep linear* NNs. Despite being sometimes regarded as too simplistic, the study of linear networks is a crucial starting point for a systematic analysis of the FA algorithm: it will allow us to rigorously analyze a tractable model and it will give us insights and intuition on the general nonlinear setting (Section 7).

Our main results are the following.

- We prove convergence of the FA continuous dynamics, with rates, for $L$-layer NNs with $L \geq 2$ (Sections 2 and 3). Such results hold for any input dimension and number of neurons: the network is not necessarily overparametrized.

- We prove that, for certain initialization schemes, the continuous FA dynamics sequentially learn the solutions of a reduced-rank regression problem, but in a backward fashion: smaller components first, dominant components later (Section 4). Additionally, we provide guidelines for an initialization scheme that avoids such phenomena.

- Finally (Section 5), we analyze the corresponding discrete FA dynamics and we prove convergence to the true labels with linear convergence rates.

We assumed some conditions on the data and on the model in order to render the exposition and the reading more clear. However, some of the "strong" assumptions that will appear in the paper can be easily relaxed (see Section 2).

## 2 WARM-UP: SHALLOW NETWORKS

In this section, we consider a 2-layer *linear* NN with vector output $\hat{\boldsymbol{y}} = \boldsymbol{W}_2\boldsymbol{W}_1\boldsymbol{x} \in \mathbb{R}^o$ and input vector $\boldsymbol{x} \in \mathbb{R}^d$ such that $(\boldsymbol{x}, \boldsymbol{y}) \sim \mathcal{D}$ for some distribution $\mathcal{D}$. Define $\boldsymbol{\Sigma}_{xx} := \mathbb{E}[\boldsymbol{x}\boldsymbol{x}^\top] \in \mathbb{R}^{d \times d}$, that we assume to be positive definite, and $\boldsymbol{\Sigma}_{xy} := \mathbb{E}[\boldsymbol{y}\boldsymbol{x}^\top] \in \mathbb{R}^{o \times d}$, then the following result holds:

**Proposition 1.** *For any distribution $\mathcal{D}$, the FA dynamics that dictates the learning process is*

$$\dot{\boldsymbol{W}}_1 = \boldsymbol{M} \left(\boldsymbol{\Sigma}_{xy} - \boldsymbol{W}_2\boldsymbol{W}_1\boldsymbol{\Sigma}_{xx}\right), \qquad \dot{\boldsymbol{W}}_2 = \left(\boldsymbol{\Sigma}_{xy} - \boldsymbol{W}_2\boldsymbol{W}_1\boldsymbol{\Sigma}_{xx}\right)\boldsymbol{W}_1^\top, \tag{3}$$

*where the dot refers to the time derivative.*

Note that this result particularly holds for empirical distributions corresponding to finite datasets.

In this work, we will assume that the two matrices $\boldsymbol{\Sigma}_{xx}, \boldsymbol{\Sigma}_{xy}$ can be simultaneously decomposed as $\boldsymbol{\Sigma}_{xx} = \boldsymbol{V}\boldsymbol{\Lambda}_{xx}\boldsymbol{V}^\top$ and $\boldsymbol{\Sigma}_{xy} = \boldsymbol{U}\boldsymbol{\Lambda}_{xy}\boldsymbol{V}^\top$, where the latter corresponds to the singular value decomposition (SVD) of $\boldsymbol{\Sigma}_{xy}$. This assumption holds in many situations, as discussed in details by Gidel et al. (2019). Moreover, a perturbation analysis similar as the one done in Gidel et al. (2019, Theorem 1) could be performed in the general case in order to handle the non-commutative case. Next, we introduce the change of variables $\boldsymbol{W}_1 = \boldsymbol{R}\tilde{\boldsymbol{W}}_1\boldsymbol{V}^\top$ and $\boldsymbol{W}_2 = \boldsymbol{U}\tilde{\boldsymbol{W}}_2\boldsymbol{R}^\top$, for some arbitrary left-orthogonal matrix $\boldsymbol{R}$, and we choose the FA matrix $\boldsymbol{M}$ to be decomposable in a similar fashion: $\boldsymbol{M} = \boldsymbol{R}\boldsymbol{D}\boldsymbol{U}^\top$, for some matrix $\boldsymbol{D}$ with positive diagonal entries $\boldsymbol{D}_{ii} > 0$ and zero entries otherwise. In particular, this implies that $\boldsymbol{M}$ is full rank, therefore guaranteeing convergence to a global minimum of the dynamics. The equations for the FA flow will then become

$$\dot{\tilde{\boldsymbol{W}}}_1 = \boldsymbol{D}\left(\boldsymbol{\Lambda}_{xy} - \tilde{\boldsymbol{W}}_2\tilde{\boldsymbol{W}}_1\boldsymbol{\Lambda}_{xx}\right) \qquad \dot{\tilde{\boldsymbol{W}}}_2 = \left(\boldsymbol{\Lambda}_{xy} - \tilde{\boldsymbol{W}}_2\tilde{\boldsymbol{W}}_1\boldsymbol{\Lambda}_{xx}\right)\tilde{\boldsymbol{W}}_1^\top. \tag{4}$$

Moreover, by the change of variable $\boldsymbol{W}_1' := \tilde{\boldsymbol{W}}_1\boldsymbol{\Lambda}_{xx}^{-1/2}$, $\boldsymbol{W}_2' := \boldsymbol{\Lambda}_{xx}^{-1/2}\tilde{\boldsymbol{W}}_2$, $\boldsymbol{D}' := \boldsymbol{D}\boldsymbol{\Lambda}_{xx}^{-1/2}$ where $\boldsymbol{\Lambda}_{xx}^{1/2}$ is the coordinate-wise square root of $\boldsymbol{\Lambda}_{xx}$,[1] we get after some simple calculations that

$$\dot{\boldsymbol{W}}_1' = \boldsymbol{D}'\left(\boldsymbol{\Lambda}_{xx}^{1/2}\boldsymbol{\Lambda}_{xy}\boldsymbol{\Lambda}_{xx}^{-1/2} - \boldsymbol{W}_2'\boldsymbol{W}_1'\right), \quad \dot{\boldsymbol{W}}_2' = \left(\boldsymbol{\Lambda}_{xx}^{1/2}\boldsymbol{\Lambda}_{xy}\boldsymbol{\Lambda}_{xx}^{-1/2} - \boldsymbol{W}_2'\boldsymbol{W}_1'\right)\boldsymbol{W}_1'^\top. \tag{5}$$

Therefore, by considering $\boldsymbol{\Lambda}_{xy}' := \boldsymbol{\Lambda}_{xx}^{1/2}\boldsymbol{\Lambda}_{xy}\boldsymbol{\Lambda}_{xx}^{-1/2}$, we may equivalently assume $\boldsymbol{\Sigma}_{xx} = \boldsymbol{I}_d$ (isotropic features), without any loss of generality.

It is now crucial to observe that if $\boldsymbol{W}_1' \in \mathbb{R}^{h \times d}, \boldsymbol{W}_2' \in \mathbb{R}^{o \times h}$ are diagonal matrices, then the dynamics decouples into $k$ independent equations with $k = \min\{d, h, o\}$. In particular, let $\theta_1^i = (\boldsymbol{W}_1')_{ii}$ and $\theta_2^i = (\boldsymbol{W}_2')_{ii}$ and assume that at time $t = 0$ the matrices $\boldsymbol{W}_1'(0)$ and $\boldsymbol{W}_2'(0)$ are diagonal. Then, for each $i = 1, \ldots, k$ we obtain the following scalar system:

$$\dot{\theta}_1^i = d^i\left(\lambda^i - \theta_2^i\theta_1^i\right) \qquad \dot{\theta}_2^i = \left(\lambda^i - \theta_2^i\theta_1^i\right)\theta_1^i, \tag{6}$$

where $\lambda^i = (\boldsymbol{\Lambda}_{xy})_{ii}$ and $d^i = \boldsymbol{D}_{ii}$. Note that for $i > k$ one could define diagonal coefficients for the matrices $\boldsymbol{W}_1'$ or $\boldsymbol{W}_2'$, but their derivative will be 0, thus non-trivial dynamics only occur for $i \in [k]$.

### 2.1 THE SCALAR DYNAMICS

We now turn our attention to analyzing a nonlinear system of the form (6) for the scalar functions $\theta_1, \theta_2 \in C^1(\mathbb{R})$ (we suppressed the indices).

**Theorem 2.** *For any FA constant $d \in \mathbb{R}_{>0}$, and $\forall \theta_0 \in \mathbb{R}$ initial value with initialization scheme*

$$\theta_1(0) = \theta_0 \qquad and \qquad \theta_2(0) = \frac{\theta_0^2}{2d}, \tag{7}$$

$\exists C > 0$ *such that the product $\theta_2(t)\theta_1(t)$ of the solution of the system* (6) *converges exponentially to the signal $\lambda$ in a monotonic fashion: $|\theta_2(t)\theta_1(t) - \lambda| < Ce^{-\frac{3}{2}(d\lambda)^{\frac{2}{3}}t}$ as $t \to +\infty$. Moreover the first layer gets "aligned" with the feedback matrix such that $\theta_1(t) \to \sqrt[3]{2rd}$.*

---

[1]Note that we used a slight abuse of notation: depending on the matrix it is multiplied to $\boldsymbol{\Lambda}_{xx}^{-1/2}$ corresponds to the diagonal matrix either of size $d \times d$ or $o \times o$ with the diagonal coefficients $[\boldsymbol{\Lambda}_{xx}^{-1/2}]_{ii}$, $1 \leq i \leq \text{rank}(\boldsymbol{\Lambda}_{xx})$.

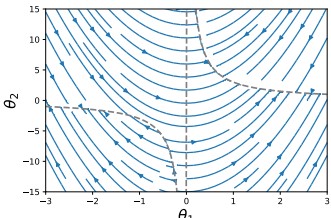
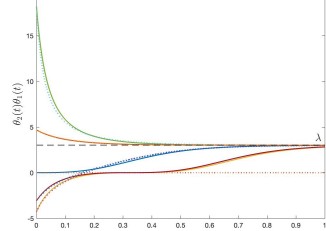
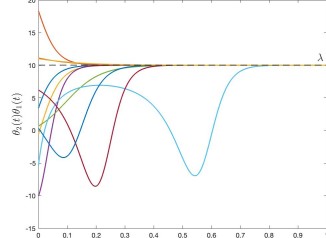

(a) FA trajectories in the phase space $(\theta_1, \theta_2)$ for $d = .2$. The dotted hyperbola correspond to the solution set $\theta_1 \theta_2 = \lambda$.

(b) FA (continuous lines) and GD (dotted lines), with initial conditions (7); $\theta_0 \sim \mathcal{U}([-5, 5])$ and $d = 2$.

(c) FA dynamics with i.i.d initial conditions $(\theta_1(0), \theta_2(0)) \sim \mathcal{U}([-5, 5]) \times \mathcal{U}([-5, 5])$ and $d = 2$.

Figure 1: Numerical simulations of the solution $\theta_2(t)\theta_1(t)$ of the FA dynamics with true signal $\lambda = 3$ (left and middle) and $\lambda = 10$ (right), with different initializations.

The proof can be found in Appendix A.2: along with the proof, it appears that the FA weights drive the learning of the forward weights for the first layers to the point that their limit is "aligned" to the value of the FA constants (in a nonlinear fashion). Similarly, this holds for the deep network case (Section 3 and Appendix B.2). One can also observe that components with a large FA constant $d$ are learned faster (see (35)–(36) in Appendix A.2), boosting the alignment.

From Theorem 2 it is clear that the choice of the FA constant will force the optimization to select *one* particular pair of solutions $(\theta_1(t), \theta_2(t))$ among the infinitely many possible solutions (the equation $\theta_1 \theta_2 = \lambda$ is a 1-dimensional manifold on the $\mathbb{R}^2$-plane). However, all of these solutions represent global minima (those are the only stationary points of FA), and therefore via the FA dynamics, we can find the global minima of the loss function.

We can easily notice that zero-initialization does guarantee convergence for FA, unlike in the case of GD dynamics (Figure 1b). Furthermore, we want to highlight the ideological novelty of the proposed initialization scheme (7): unlike considering all weight matrices as independent players at the beginning of the optimization process and initializing them at random, we are imposing that once the weight matrix $W_1$ of the first layers is initialized, the initial value of the weight matrix $W_2$ of the second layer is deterministic, and it depends directly from the first layer.

More generally, it is easy (though lengthy) to prove that the product of the solutions $(\theta_1(t), \theta_2(t))$ of the system (6) always converges to the signal $\lambda$, regardless of the initialization scheme considered. Some intuitions can be built from the phase diagram in Fig 1a where one can see that the trajectories of the dynamics lie on parabolas with equation $2d\theta_2 = \theta_1^2 + 2dK$ while the solution set is represented by the hyperbola $\theta_1 \theta_2 = \lambda$. The difficulties in the analysis arise when the initialization correspond to a parabola that intersect the solution set more than once. Moreover, the non-monoticity of the quantity $|\theta_1(t)\theta_2(t) - \lambda|$ can be seen on this diagram as the result of some trajectories getting close to the bottom left part of the hyperbola but eventually converging to the top right one.

**Theorem 3.** *For any FA constant $d \in \mathbb{R}_{>0}$, and for any initial values $\theta_1(0)$ and $\theta_2(0)$, the system (6) admits a unique solution $(\theta_1(t), \theta_2(t))$ such that $\theta_1(t), \theta_2(t)$ are bounded for all times $t \geq 0$ and $\theta_2(t)\theta_1(t) \to \lambda$ as $t \to +\infty$. The convergence rate is exponential for any pair of initial values $(\theta_1(0), \theta_2(0))$ except for a set of (Lebesgue) measure zero in $\mathbb{R}^2$.*

For the proof, see Appendix A.3. The main consequence of Theorem 3 is that even by initializing all layers randomly and independently (as it is common practice), the FA dynamics still converges to the true signal. However, we want to stress that the initialization scheme (7) is more suitable for ML applications: for arbitrary initial conditions, the product of the solutions $\theta_2(t)\theta_1(t)$ may not be monotonic (compare with Theorem 2). Thus it may trigger inefficient early stopping practices (see Figure 1c). Furthermore, for a randomly and independently sampled pair of initial conditions, we may observe a highly undesirable "reversed" incremental learning phenomenon (Section 4), which is avoided if (7) is put into place.

**Relaxing some assumptions.** Considering the the assumption that the weight matrices $W_1, W_2$ share the left/right singular vectors with $\Sigma_{xy}$ is not strictly needed: with more work, the more general case can be analyzed via a perturbation analysis, along the same lines as in Gidel et al. (2019) (Assumption 1). Furthermore, even though the initialization of the matrices $\tilde{W}_1, \tilde{W}_2$ are diagonal

Saxe et al. (2018) argues that the diagonal solutions provide good approximations to the full solutions when $\tilde{W}_1(0), \tilde{W}_2(0)$ are full matrices, initialized with small random weights.

## 3 DEEP NETWORKS

In this section we extend the results of Section 2 to deep linear NNs: given a distribution of input-output pairs $(\boldsymbol{x}, \boldsymbol{y}) \sim \mathcal{D}$, the predicted output vector is now $\hat{\boldsymbol{y}} = \boldsymbol{W}_L \ldots \boldsymbol{W}_2 \boldsymbol{W}_1 \boldsymbol{x}$ with weight matrices $\boldsymbol{W}_\ell \in \mathbb{R}^{h_\ell \times h_{\ell-1}}$ ($h_0 = d$, $h_L = o$) and we note $\boldsymbol{\Sigma}_{xx} = \mathbb{E}[\boldsymbol{x}\boldsymbol{x}^\top] \in \mathbb{R}^{d \times d}$ and $\boldsymbol{\Sigma}_{xy} = \mathbb{E}[\boldsymbol{y}\boldsymbol{x}^\top] \in \mathbb{R}^{o \times d}$.

**Proposition 4.** *For any distribution $\mathcal{D}$, the FA equations of motion are*

$$\dot{\boldsymbol{W}}_\ell = \boldsymbol{M}_\ell(\boldsymbol{\Sigma}_{xy} - \boldsymbol{W}_{[1:L]}\Sigma_{xx})\boldsymbol{W}_{[1:\ell-1]}^\top \qquad \ell \in [L], \tag{8}$$

*for some FA weight matrices $\{\boldsymbol{M}_\ell\}_{\ell \in [L]}$, with $\boldsymbol{M}_L = \boldsymbol{I}$, and with $\boldsymbol{W}_{[1:\ell]} = \boldsymbol{W}_\ell \ldots \boldsymbol{W}_1$.*

Notice that the legitimate FA update would be $\dot{\boldsymbol{W}}_\ell = \tilde{\boldsymbol{M}}_{[\ell:L-1]}(\boldsymbol{\Sigma}_{xy} - \boldsymbol{W}_{[1:L]}\Sigma_{xx})\boldsymbol{W}_{[1:\ell-1]}^\top$, however since we are requiring the FA matrices to be full rank, we can redefine the FA matrices as $\boldsymbol{M}_\ell = \tilde{\boldsymbol{M}}_{[\ell:L-1]}$. We now perform a similar change of variables as in Section 2: $\boldsymbol{W}_\ell = \boldsymbol{R}_\ell \tilde{\boldsymbol{W}}_\ell \boldsymbol{R}_{\ell-1}^\top$ with $\boldsymbol{R}_0 = \boldsymbol{V}, \boldsymbol{R}_L = \boldsymbol{U}$ and $\{\boldsymbol{R}_\ell\}_{\ell=1,\ldots,L-1}$ arbitrary left-orthogonal matrices. We additionally assume that the FA matrices $\boldsymbol{M}_\ell$ have the following prescribed form $\boldsymbol{M}_\ell = \boldsymbol{R}_\ell \boldsymbol{D}_\ell \boldsymbol{U}^\top$ (with $\boldsymbol{D}_\ell$ a matrix with zero entries except for entries on the diagonal, which are positive). Then, the system becomes

$$\dot{\tilde{\boldsymbol{W}}}_\ell = \boldsymbol{D}_\ell(\boldsymbol{\Lambda}_{xy} - \tilde{\boldsymbol{W}}_{[1:L]}\boldsymbol{\Lambda}_{xx})\tilde{\boldsymbol{W}}_{[1:\ell-1]}^\top \qquad \ell \in [L]. \tag{9}$$

Considering the change of variable $\boldsymbol{W}_1' := \tilde{\boldsymbol{W}}_1 \boldsymbol{\Lambda}_{xx}^{-1/2}$, $\boldsymbol{W}_L' := \boldsymbol{\Lambda}_{xx}^{-1/2}\tilde{\boldsymbol{W}}_L$, $\boldsymbol{D}_\ell' := \boldsymbol{D}_\ell \boldsymbol{\Lambda}_{xx}^{-1/2}$, $\boldsymbol{\Lambda}_{xy}' := \boldsymbol{\Lambda}_{xx}^{1/2}\boldsymbol{\Lambda}_{xy}\boldsymbol{\Lambda}_{xx}^{-1/2}$ and assuming as in §2 that $\boldsymbol{W}_j'(0)$ are diagonal, the system decouples:

$$\dot{\theta}_\ell^i = d_\ell^i(\lambda^i - \theta_L^i \ldots \theta_1^i)\theta_{\ell-1}^i \ldots \theta_1^i, \qquad \ell \in [L], \tag{10}$$

and for each entry $i = 1, \ldots, k$, with $k = \min\{d, h_1, \ldots, h_{L-1}, o\}$, the analysis reduces to studying a system of scalar functions $\{\theta_1^i, \ldots, \theta_L^i\}$.

**Theorem 5.** *For any set of FA constants $\{d_\ell\}_{\ell \in [L-1]}$, $d_\ell \in \mathbb{R}_{>0}$ and for any $\theta_0 \in \mathbb{R}$, there exists an initialization scheme of the form*

$$\theta_1(0) = \theta_0, \qquad \theta_\ell(0) = C_\ell \theta_0^{2^{\ell-1}}, \qquad \ell = 2, \ldots, L, \tag{11}$$

*where $C_\ell \in \mathbb{R}_{>0}$ depends on $\{d_1, \ldots, d_\ell\}$, such that the system of differential equations (10) has a unique solution $(\theta_1(t), \ldots, \theta_L(t))$, which converges exponentially to the signal $\lambda$ as $t \to +\infty$ in a monotone fashion.*

As in the 2-layer case, the proposed initialization scheme is characterized by the fact that only the first layer is initialized at random, while all the other layers depend deterministically on it. Note that, along the same lines as Theorem 3, it is also possible to prove convergence to the true signal for any initialization set $\{\theta_\ell(0)\}_{\ell \in [L]}$. However, some desirable properties of the dynamics (monotonicity, exponential convergence) may be lost. For the proofs and the explicit construction of the initialization scheme (11) see Appendix B.2.

## 4 IMPLICIT REGULARIZATION PHENOMENA

In this section, we analyze the phenomenon of hierarchical learning (or implicit regularization) for FA, as described in Gidel et al. (2019) and Gissin et al. (2019) in the case of GD: along the optimization path, features of the model are learned in a sequential way, so that one feature is fully learned before the dynamics starts to learn the next feature. For FA, we may observe a similar behavior as GD, however, in certain settings the dynamics seems to favor the learning of secondary component prior than the learning of principal components, i.e. signals $\lambda^i$'s with small magnitude are learned earlier than those of big magnitude. This phenomenon is potentially highly problematic since smaller singular values $\lambda^i$'s are normally tied to noisy measurements and do not represent nor

give meaningful information on the structure of the data. Privileging negligible features goes against the standard principles of reasoned data analysis (Principal Component Analysis, for example). We will see that the occurrence of this behavior depends on particular initializations of the algorithm and it can be therefore successfully avoided.

**Theorem 6** (Informal). *Let $\lambda_1 > \ldots > \lambda^k$ be the principal components of $\mathbf{\Lambda}_{xy}$. There exists initialization schemes such that the component $\lambda_i$ is incrementally learned at time $T_i$. Moreover, there exists two different initialization schemes such that*

$$\textit{Anti-Regularization(§4.1):} \ \ T_1 > \ldots > T_k \quad \textit{and} \quad \textit{Regularization(§4.2):} \ \ T_1 < \ldots < T_k \quad (12)$$

To illustrate such a phenomenon and rigorously describe it, consider the 2-layer case in the SVD-setting as in Section 2: $\forall \, i = 1, \ldots, k$, the system (6) can be equivalently rewritten as

$$\dot{\theta}_1^i = -\frac{1}{2} \left( (\theta_1^i)^3 + 2dK^i \theta_1^i - 2d\lambda^i \right), \qquad \theta_1^i(0) = \theta_0^i, \tag{13}$$

and $\theta_2^i(t) = \frac{1}{2d} \theta_1^i(t)^2 + K^i$, with $K^i = \theta_2^i(0) - \frac{1}{2d^i}(\theta_1^i(0))^2 \in \mathbb{R}$ and $d, \lambda^i > 0$ (for simplicity we assumed $d^i = d, \forall \, i$). Depending on the value of $K^i$ (i.e. the initial condition of the $i$-th system), the cubic polynomial on the right-hand side of (13) may have between 1 and 3 distinct real roots. The value of the discriminant $\Delta = -4d^2 \left( 8d(K^i)^3 + 27(\lambda^i)^2 \right)$ will dictate which regime we fall into.

We will be considering the case where $\Delta > 0$ and the case $K^i = 0$ (i.e. the initialization scheme (7)), which is a special instance of the case $\Delta < 0$. The case $\Delta = 0$ can be equally analyzed, but it will not be considered in this paper, as it has zero probability of occurring.[2] The sequential learning phenomenon becomes evident when we perform a double scaling limit of the time variable $t \mapsto \delta t$ and the initial condition $\theta_0^i \mapsto f(\delta, \theta_0^i), \delta \to +\infty$, as it will be clear below. Note that the rescaling is uniform across the components $i$. Proof of Theorem 7 and lengthier discussion can be found in Appendix C.

## 4.1 Case $\Delta > 0$: backward learning

If $K^i < -\frac{3}{2}(\lambda^i)^{3/2} < 0$, the cubic polynomial in (13) has 3 real, distinct root $r_1^i < r_2^i < r_3^i$. The intuition is that by initializing $\theta_0^i$ at a value that is very close to $r_2^i$ and applying the time scaling $t \mapsto \delta t, \delta \gg 1$ (i.e. we are "fast-forwarding" the solution $\theta_1(\delta t)$), we can clearly identify the time threshold $T^i$ after which the signal is fully learned. More formally,

**Theorem 7.** $\forall \, i = 1, \ldots, k$, if $\theta_1^i(0) = r_2^i + e^{-\delta}$ ($\delta > 0$), then the solution $\theta_1^i(\delta t)$ converges to a step function:

$$\theta_1^i(\delta t) \to r_2^i \mathbb{I}_{\{t < T^i\}} + \alpha^i \mathbb{I}_{\{t = T^i\}} + r_3^i \mathbb{I}_{\{t > T^i\}} \qquad as \ \delta \to +\infty, \tag{14}$$

where $\mathbb{I}_{\{t \in A\}}$ is the indicator function on the set A,

$$T^i = \frac{2}{(r_3^i - r_2^i)(r_2^i - r_1^i)}, \qquad \alpha^i \in (r_2^i, r_3^i). \tag{15}$$

*Furthermore, $T^i$ is an increasing function of $\lambda^i$: for $\lambda^1 > \lambda^2 > \ldots > \lambda^k$ and fixed FA constant $d$, we have $T^1 > T^2 > \ldots > T^k$.*

Similar conclusions hold when $\theta_1^i(0) = r_2^i - e^{-\delta}$. Therefore, as the value of $\lambda^i$ increases, the value of $T^i$ increases as well: negligible features of the data are learned sooner than the dominant ones.

On the other hand, Theorem 7 shows that such a backward incremental learning does *not* happen with high probability: indeed, we are requiring all the parameters $K^i$ (which depends on the initialization) to be sufficiently negative and the probability of sampling initial conditions that will yield anti-regularization depends on the value of the signals $\lambda^i$ (i.e., it is data-dependent): the bigger the value of $\lambda^i$, the smaller the probability. Nevertheless, we will see in the next subsection how such a phenomenon can be safely bypassed and we discuss some properties that suggest that the initialization scheme (7) can guarantee the correct sequential learning of features as GD exhibits.

---

[2]The claim is correct if we sample the initial conditions $\theta_1(0), \theta_2(0)$ independently and generically: for example, from a uniform distribution $\mathcal{U}([-R, R]) \times \mathcal{U}([-R, R])$, for some $R \in \mathbb{R}_{>0}$.

## 4.2 CASE $K = 0$: EVIDENCE OF INCREMENTAL LEARNING

Imposing $K^i = 0$ for all $i$'s means to recur to the initialization scheme (7): $\theta_1^i(0) = \theta_0^i$, $\theta_2^i(0) = \frac{1}{2d}(\theta_0^i)^2$. In this case, Theorem 2 already provides exponential convergence rates that depend on the magnitude of the signal $(|\theta_2^i(t)\theta_1^i(t) - \lambda^i| < Ce^{-\frac{3}{2}(d\lambda^i)^{\frac{2}{3}}t}$ as $t \to +\infty)$, suggesting that the bigger the magnitude of the signal $\lambda^i$, the faster the convergence.

If $\theta_0^i < 0$, there exists a unique time $T_0^i$ (vanishing time) such that $\theta_1^i(T_0^i)\theta_2^i(T_0^i) = 0$ and $\theta_1^i(t)\theta_2^i(t) \lessgtr 0$ for $t \lessgtr T_0^i$ respectively, where

$$T_0^i = \frac{\pi}{(r^i)^2 3\sqrt{3}} + \frac{1}{3(r^i)^2} \ln\left(\frac{(r^i - \theta_0^i)^2}{(\theta_0^i)^2 + r^i\theta_0^i + (r^i)^2}\right) - \frac{2}{(r^i)^2\sqrt{3}} \arctan\left(\frac{2\theta_0^i + r^i}{r^i\sqrt{3}}\right). \quad (16)$$

We can interpret such a vanishing time as the first milestone of the learning process. In the regime as $\theta_0^i \to -\infty \; \forall \, i = 1, \ldots, k$, we have

$$T_0^i = \frac{4\pi}{3(r^i)^2\sqrt{3}} + \mathcal{O}\left(\frac{1}{(\theta_0^i)^2}\right); \quad (17)$$

where $r^i = \sqrt[3]{2d\lambda^i}$, therefore for bigger $\lambda^i$'s (i.e. dominant singular values) the vanishing time happens sooner: for $\lambda^1 > \lambda^2 > \ldots > \lambda^k$ and fixed FA constant $d > 0$, $T_0^1 < T_0^2 < \ldots < T_0^k$.

## 5 DISCRETE DYNAMICS

### 5.1 FORWARD EULER DISCRETIZATION

In this section, we are interested in analyzing the discrete FA dynamics and its convergence properties. For the discrete time setting, we focus again on a 2-layer linear NN. The standard forward Euler method yields the following discretized system:

$$\boldsymbol{W}_1^{(t+1)} = \boldsymbol{W}_1^{(t)} + \eta \boldsymbol{M}\left(\boldsymbol{\Sigma}_{xy} - \boldsymbol{W}_2^{(t)}\boldsymbol{W}_1^{(t)}\right) \quad (18)$$

$$\boldsymbol{W}_2^{(t+1)} = \boldsymbol{W}_2^{(t+1)} + \eta\left(\boldsymbol{\Sigma}_{xy} - \boldsymbol{W}_2^{(t)}\boldsymbol{W}_1^{(t)}\right)\left(\boldsymbol{W}_1^{(t)}\right)^\top \quad (19)$$

and performing again the SVD change of variables, the system becomes

$$\tilde{\boldsymbol{W}}_1^{(t+1)} = \tilde{\boldsymbol{W}}_1^{(t)} + \eta \boldsymbol{D}\left(\boldsymbol{\Lambda}_{xy} - \tilde{\boldsymbol{W}}_2^{(t)}\tilde{\boldsymbol{W}}_1^{(t)}\right) \quad (20)$$

$$\tilde{\boldsymbol{W}}_2^{(t+1)} = \tilde{\boldsymbol{W}}_2^{(t)} + \eta\left(\boldsymbol{\Lambda}_{xy} - \tilde{\boldsymbol{W}}_2^{(t)}\tilde{\boldsymbol{W}}_1^{(t)}\right)\left(\tilde{\boldsymbol{W}}_1^{(t)}\right)^\top \quad (21)$$

If we set the initial conditions $\boldsymbol{W}_1^{(0)}$, $\boldsymbol{W}_2^{(0)}$ as diagonal, the matrices remain diagonal for all times and the dynamic decouples. For each $i = 1, \ldots, k$, we have

$$\left(\theta_1^{(t+1)}\right)^i = \left(\theta_1^{(t)}\right)^i + \eta d^i\left(\lambda^i - \left(\theta_2^{(t)}\right)^i\left(\theta_1^{(t)}\right)^i\right) \quad (22)$$

$$\left(\theta_2^{(t+1)}\right)^i = \left(\theta_2^{(t)}\right)^i + \eta\left(\lambda^i - \left(\theta_2^{(t)}\right)^i\left(\theta_1^{(t)}\right)^i\right)\left(\theta_1^{(t)}\right)^i \quad (23)$$

The following result holds (the index $i$ is suppressed for simplicity).

**Theorem 8.** *Consider the system* (22)–(23). *With zero initialization* $\theta_1^{(0)} = \theta_2^{(0)} = 0$, $\forall \lambda, d > 0$ *and with constant step size* $\eta$ *small enough, the product* $\theta_2^{(t)}\theta_1^{(t)}$ *converges to the true signal* $\lambda$ *with linear rates:* $\left|\theta_2^{(t)}\theta_1^{(t)} - \lambda\right| \leq Cq^t$, *for some* $C > 0$.

For the sake of clarity, we deferred the precise prescription for the magnitude of $\eta$ and the linear rate $q$ to Appendix D, where the full proof of the Theorem can be found.

## 5.2 MODIFIED FORWARD DISCRETIZATION

In addition to the standard discretization strategy, we are proposing here a new discretization technique that is particularly instrumental for the FA setting, as it will allow a thorough inspection of its convergence guarantees and convergence rates for NNs of any depth. Inspired by the fact that the behaviour of the second layer strictly depend on the choices that we impose on the first layer in the continuous dynamics (Theorem 2 and initialization (7)), we slightly modify the update rule for second weight matrix $W_2$ in the following way:

$$W_1^{(t+1)} = W_1^{(t)} + \eta M \left( \Sigma_{xy} - W_2^{(t)} W_1^{(t)} \right) \tag{24}$$

$$W_2^{(t+1)} = W_2^{(t+1)} + \eta \left( \Sigma_{xy} - W_2^{(t)} W_1^{(t)} \right) \left( W_1^{\left(t+\frac{1}{2}\right)} \right)^\top \tag{25}$$

with $W_1^{\left(t+\frac{1}{2}\right)} = \frac{1}{2} \left( W_1^{(t+1)} + W_1^{(t)} \right)$. The above discretization technique can be considered as a hybrid between the Euler forward method and the trapezoidal rule (Iserles, 1996), although we are still evaluating the error $\Sigma_{xy} - W_2^{(t)} W_1^{(t)}$ at the present time $t$. The underlying idea is that we want to already use the information encoded in the new $(t+1)$-iterate of $W_1$ to update the second layer by averaging $W_1^{(t)}$ and $W_1^{(t+1)}$.

Performing again the SVD change of variables, and setting $W_1^{(0)}, W_2^{(0)}$ as diagonal, we obtain a scalar system of the form ($\forall\ i = 1, \dots, k$)

$$\left(\theta_1^i\right)^{(t+1)} = \left(\theta_1^i\right)^{(t)} + \eta d^i \left( \lambda^i - \left(\theta_2^i\right)^{(t)} \left(\theta_1^i\right)^{(t)} \right) \tag{26}$$

$$\left(\theta_2^i\right)^{(t+1)} = \left(\theta_2^i\right)^{(t)} + \eta \left( \lambda^i - \left(\theta_2^i\right)^{(t)} \left(\theta_1^i\right)^{(t)} \right) \left(\theta_1^i\right)^{\left(t+\frac{1}{2}\right)}. \tag{27}$$

A convergence result follows (suppressing the $i$-dependence).

**Theorem 9.** *Consider the system* (26)–(27). *With zero initialization* $\theta_1^{(0)} = \theta_2^{(0)} = 0, \forall\ \lambda, d > 0$ *and with constant step size* $\eta < \frac{2}{3(2d\lambda)^{\frac{2}{3}}}$, *the product* $\theta_2^{(t)} \theta_1^{(t)}$ *linearly converges to the true signal* $\lambda$: $|\theta_2^{(t)} \theta_1^{(t)} - \lambda| \leq 3\lambda(1 - \frac{3\eta}{2}(2d\lambda)^{\frac{2}{3}})^t$.

The discretization scheme easily generalizes to the case of $L$ layers for $L \geq 3$ (we again suppress the $i$-index for readability):

$$\theta_\ell^{(t+1)} = \theta_\ell^{(t)} + \eta d_\ell \theta_{[1:\ell-1]}^{\left(t+\frac{1}{2}\right)} \left( \lambda - \theta_L^{(t)} \dots \theta_1^{(t)} \right), \qquad \ell = 1, \dots, L, \tag{28}$$

with $d_L = 1$. Also in this case, the dynamics closely mimic the continuous counterpart, reducing the system to a single recurrence equation for $\{\theta_1^{(t)}\}$. By selecting a step size $\eta$ small enough, the sequence of products $\{\theta_L^{(t)} \dots \theta_2^{(t)} \theta_1^{(t)}\}$ will converge to the signal $\lambda$ with exponential rate.

**Theorem 10.** *Consider the system* (28) *with* $\lambda, d_\ell > 0$ *and zero initialization* $\theta_\ell^{(0)} = 0, \forall\ \ell = 1, \dots, L$. *If the constant step size satisfies* $\eta < (d_1 \gamma (\mathfrak{K}\lambda^{\gamma-1})^{1/\gamma})^{-1}$, *where* $\gamma = 2^L - 1$ *and* $\mathfrak{K}$ *is a suitable positive constant depending on the FA constants* $d_1, \dots, d_{L-1}$, *then the product* $\prod_{\ell=1}^L \theta_\ell^{(t)}$ *linearly converges to the true signal* $\lambda$: $|\theta_L^{(t)} \dots \theta_1^{(t)} - \lambda| \leq C_\lambda (1 - \eta d_1 \gamma (\mathfrak{K}\lambda^{\gamma-1})^{\frac{1}{\gamma}})^t$, *for some constant* $C_\lambda \in \mathbb{R}_{>0}$ *only dependent on* $\lambda$.

The proofs of Theorems 9 and 10 can be found in Appendix D.

## 6 EXPERIMENTS

In this last section, we discuss one ML application of our theoretical results. The set-up is borrowed from Gidel et al. (2019) (Section 4.2), where a linear autoencoder is considered.

For an autoencoder, the (true) output is equal to its input $y = x \in \mathbb{R}^d$ and we want to compare the reconstruction properties of $W_2^{(t)} W_1^{(t)}$ (2 layer NN) and $W_3^{(t)} W_2^{(t)} W_1^{(t)}$ (3 layer NN) computed via

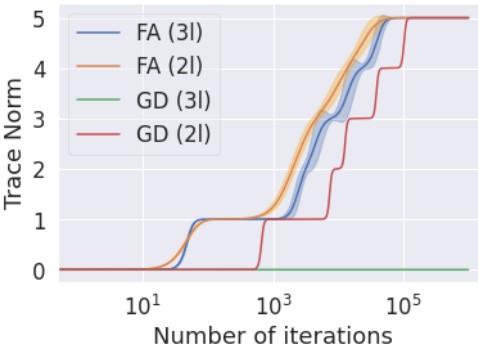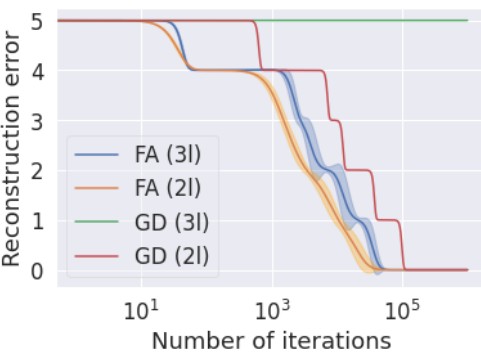

Figure 2: Comparison of trace norms and reconstruction errors for linear autoencoders. For the FA experiments, we are plotting the average (with error bars) of 15 experiments.

FA versus GD. In this experiment, we have $d = o = 20$ and $k = 5$. We generate $n = 1000$ synthetic data $\{x_i\}$ in the following way: each data point is given as $x_i = Az_i + \epsilon_i$, where $A \in \mathbb{R}^{d \times k}$ is a fixed matrix with entries sampled as $A_{kl} \sim \mathcal{U}([0,1])$, $z_i \sim \mathcal{N}\left(0, \Lambda := \operatorname{diag}\left\{4, 2, 1, \frac{1}{2}, \frac{1}{4}\right\}\right)$ and the noise $\epsilon_i \sim 10^{-3}\mathcal{N}(0, I_d)$. We initialize the weight matrices to a close-to-zero initialization $W_1^{(0)}, W_2^{(0)}, W_3^{(0)} \sim 10^{-5}\mathcal{U}([0,1])$: this will guarantees the FA dynamics to avoid the anti-regularization pattern and to converge to the signal (Theorems 8, 9 and 10).
We generate the FA matrices $M$ (2-layers) and $M_1, M_2$ (3-layers) with entries $M_{ij} \sim \mathcal{U}([0,1])$. Keeping the same initial conditions $W_\ell^{(0)}$ and same fixed matrix $A$, we repeat the FA training 15 times, sampling a different set of FA matrices each time: in Figure 2, we report the average (with error bars) of the trace norm $\|\Theta^{(t)}\|_*$ with $\Theta^{(t)} := W_2^{(t)}W_1^{(t)}$ or $\Theta^{(t)} := W_3^{(t)}W_2^{(t)}W_1^{(t)}$ respectively, as well as thee average of the reconstruction errors $\|\Theta^{(t)} - A\Lambda A^\top\|_2$, as a function of $t$ the number of iterations.

We can see that the FA learning dynamics is distinctly faster than GD, which is learning at a much slower pace, as we initialized the dynamics close to one of GD's local maxima. In particular, the 3-layer NN trained with GD hasn't yet started to learn at the end of the selected training period of the experiment. Notably, the FA training for the 3-layer NN hints at the presence of an implicit regularization behavior, similar to what has been observed and proved in Gidel et al. (2019) for a 2-layer NN. Additional technical details can be found in Appendix E.

## 7 FINAL DISCUSSIONS AND INSIGHTS

In the present paper we rigorously analyze a biologically plausible optimization algorithm, we describe its potential as well as its limitations in possible ML applications. We provide convergence guarantees in both the continuous and discrete-time settings; additionally, we thoroughly discuss implicit regularization phenomena that may arise with certain initialization schemes and that may deeply affect the optimization effectiveness. The occurrence of drastically opposite regularization by only modifying an algorithm's initialization showcases the critical importance of the latter for ML applications.

Although the analysis is conducted only for linear models, the results give already numerous insights on the behaviour of nonlinear architectures. The analysis of Theorem 5 shows that, in some regimes, the deeper layers behave as "a power" of the first layer: this result suggests that each layer is converging at a different speed. It thus defines a hierarchy (in terms of speed of convergence) between the layers of the network. Furthermore, one can notice that in the linear case FA "removes" some stationary points: unlike GD that potentially has many stationary points that are saddle points, all the stationary points of FA are global minima. Therefore, we believe that FA will also "remove" some undesirable stationary points in the non-linear case. Several challenges remain, but the above pieces of intuition will be critical for the analysis in the nonlinear setting.

Beyond this, it is enticing to conduct a comparison between FA and GD and illustrate the advantages of one algorithm over the other (and in which setting). All of these directions will be the central objects of study in future works.

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

# A CONVERGENCE OF FA FOR 2-LAYER NNS

## A.1 PROOF OF PROPOSITION 1

*Proof.* We have that $\mathcal{L} := \frac{1}{2}\mathbb{E}[\|\boldsymbol{y} - \hat{\boldsymbol{y}}\|^2]$ and $\hat{\boldsymbol{y}} = \boldsymbol{W}_2\boldsymbol{W}_1\boldsymbol{x}$, thus

$$\nabla_{W_1}\mathcal{L} = \mathbb{E}[\tfrac{1}{2}\nabla_{W_1}\|\boldsymbol{y} - \hat{\boldsymbol{y}}\|^2] \tag{29}$$

$$= \mathbb{E}[\boldsymbol{W}_2^\top(\boldsymbol{y} - \boldsymbol{W}_2\boldsymbol{W}_1\boldsymbol{x})\boldsymbol{x}^\top] \tag{30}$$

$$= \boldsymbol{W}_2^\top(\boldsymbol{\Sigma}_{xy} - \boldsymbol{W}_2\boldsymbol{W}_1\boldsymbol{\Sigma}_{xx}), \tag{31}$$

where the last line comes from the linearity of the expectation. Similarly we have,

$$\nabla_{W_2}\mathcal{L} = \mathbb{E}[\tfrac{1}{2}\nabla_{W_2}\|\boldsymbol{y} - \hat{\boldsymbol{y}}\|^2] \tag{32}$$

$$= \mathbb{E}[(\boldsymbol{y} - \boldsymbol{W}_2\boldsymbol{W}_1\boldsymbol{x})\boldsymbol{x}^\top\boldsymbol{W}_1^\top] \tag{33}$$

$$= (\boldsymbol{\Sigma}_{xy} - \boldsymbol{W}_2\boldsymbol{W}_1\boldsymbol{\Sigma}_{xx})\boldsymbol{W}_1^\top. \tag{34}$$

Now we conclude by noticing that by definition of the FA dynamics (2), the matrix $\boldsymbol{W}_2^\top$ is replaced by a random matrix $\boldsymbol{M}$. □

## A.2 PROOF OF THEOREM 2

Consider the following system of ODEs for the functions $\theta_1, \theta_2 \in C^1(\mathbb{R})$:

$$\dot{\theta}_1 = d\,(\lambda - \theta_2\theta_1) \tag{35}$$

$$\dot{\theta}_2 = (\lambda - \theta_2\theta_1)\,\theta_1 \tag{36}$$

for some $d, \lambda \in \mathbb{R}_{>0}$. From equation (35), we substitute $\frac{\dot{\theta}_1}{d} = (\lambda - \theta_2\theta_1)$ in the second equation (36) to obtain

$$\dot{\theta}_2 = \frac{1}{2d}\dot{\theta}_1^2, \qquad \text{i.e.} \quad \theta_2(t) = \frac{1}{2d}\theta_1^2(t) + K, \tag{37}$$

for some integration constant $K \in \mathbb{R}$. We then substitute such expression for $\theta_2(t)$ back into the equation (35) in order to obtain a closed form expression for $\dot{\theta}_1$:

$$\dot{\theta}_1 = d\left(\lambda - \frac{1}{2d}\theta_1^3 - K\theta_1\right) \tag{38}$$

We now choose the initial conditions for $\theta_1(t)$ and $\theta_2(t)$ in the following way:

$$\theta_1(0) = \theta_0 \qquad \text{and} \qquad \theta_2(0) = \frac{\theta_0^2}{2d}, \tag{39}$$

so that the equation for $\theta_1$ becomes

$$\dot{\theta}_1 = \frac{1}{2}\left(2d\lambda - \theta_1^3\right) \tag{40}$$

We are now ready to prove Theorem 2, which we recall here for convenience:

**Theorem 11.** *For any FA constant $d \in \mathbb{R}_{>0}$, and $\forall\,\theta_0 \in \mathbb{R}$ initial value with initialization scheme (39), $\exists\,C > 0$ such that the product $\theta_2(t)\theta_1(t)$ of the solution of the system (35)–(36) converges exponentially to the signal $\lambda$ and in a monotonic fashion:*

$$|\theta_2(t)\theta_1(t) - \lambda| < Ce^{-\frac{3}{2}(d\lambda)^{\frac{2}{3}}t} \qquad \text{as } t \to +\infty. \tag{41}$$

*Proof.* By Picard–Lindelöf Theorem, equation (40) admits a unique local solution $\theta_1 \in C^1([0, T])$ ($T > 0$), for any initial condition $\theta_0 \in \mathbb{R}$. The local solution can be then easily extended to a global solution on $[0, +\infty)$.

The cubic polynomial on the right hand side of (40) has a unique real root $r := \sqrt[3]{2d\lambda}$: $2d\lambda - \theta_1^3 = (r - \theta_1-)\left(\theta_1^2 + \theta_1 r + r^2\right)$ where the quadratic $\theta_1^2 + \theta_1 r + r^2$ is irreducible (indeed, its discriminant is $\Delta = -3r^2 < 0$).

If $\theta_0 = r$, the solution of the ODE (40) is the constant function $\theta_1(t) = r, \forall\, t \in \mathbb{R}_{\geq 0}$. In this case, it is trivial to see that $\theta_2(t) = \frac{r^2}{2d}$ and $\theta_2(t)\theta_1(t) = \lambda, \forall\, t \in \mathbb{R}_{\geq 0}$.

If $\theta_0 \neq r$, we simply need to analyze a separable ODE whose solution reads

$$-\frac{2}{3r^2}\ln|\theta_1(t) - r| + \frac{1}{3r^2}\ln\left(\theta_1^2(t) + r\theta_1(t) + r^2\right) + \frac{2}{r^2\sqrt{3}}\arctan\left(\frac{2\theta_1(t) + r}{r\sqrt{3}}\right) + C_0 = t \tag{42}$$

with constant of integration

$$C_0 = \frac{2}{3r^2}\ln|\theta_0 - r| - \frac{1}{3r^2}\ln\left|\theta_0^2 + r\theta_0 + r^2\right| - \frac{2}{r^2\sqrt{3}}\arctan\left(\frac{2\theta_0 + r}{r\sqrt{3}}\right). \tag{43}$$

Equation (42) is transcendental and it cannot be solved for $\theta_1$ explicitly. However, we can still analyze the behaviour of the solution.

**Lemma 12.** $\theta_1(t)$ *is monotonic: in particular,* $\dot{\theta}_1 < 0$*, if* $\theta_0 > r$*, and* $\dot{\theta}_1 > 0$*, if* $\theta_0 < r$*.*

The above claims easily follows by inspecting the sign of the right hand side of (40). In particular, Lemma 12 implies that $\theta_1(t)$ is always bounded, for any initial condition $\theta_0 \in \mathbb{R}$.

As $t \to +\infty$, the left hand side of (42) may only diverge in the first logarithmic term ($\theta_1(t) \to r$), since the term involving the arctangent is always bounded and the other logarithmic term has argument that is always bounded (Proposition 12) and positive (the quadratic is irreducible).

Therefore, as $t \to +\infty$

$$\theta_1(t) \to r \qquad \text{and} \qquad \theta_2(t)\theta_1(t) \to \frac{1}{2d}r^3 = \lambda \tag{44}$$

for any initial condition $\theta_0 \in \mathbb{R}$. See Figure 3.

Furthermore, we can derive from (42) the rate of convergence:

$$|\theta_1(t) - r| =$$

$$\exp\left\{-\frac{3r^2}{2}t + \underbrace{\frac{3r^2 C_0}{2} + \frac{1}{2}\ln|\theta_1(t)^2 + r\theta_1(t) + r^2| + \sqrt{3}\,\text{sign}(r)\arctan\left(\frac{2\theta_1(t) + r}{|r|\sqrt{3}}\right)}_{=:g(t)}\right\} \tag{45}$$

where $g(t) = \frac{3r^2 C_0}{2} + \frac{1}{2}\ln(3r^2) + \frac{\pi}{\sqrt{3}} + o(1)$, as $t \to \infty$; therefore,

$$|\theta_1 - r| = e^{-\frac{3r^2}{2}t + \mathcal{O}(1)}. \tag{46}$$

Finally, from $\theta_2(t) = \frac{1}{2d}\theta_1(t)^2$, we get the desired result:

$$\left|\theta_2(t)\theta_1(t) - \frac{1}{2d}r^3\right| = |\theta_2(t)\theta_1(t) - \lambda| \leq Ce^{-\frac{3r^2}{2}t} \tag{47}$$

for some positive constant $C$.

**Remark 13.** *All the above calculations are valid also in the general case* $r, \lambda \in \mathbb{R} \setminus \{0\}$*. In the case where* $r < 0$ *(i.e.* $\text{sign}(d) \neq \text{sign}(\lambda)$*), the solution* $\theta_1(t)$ *satisfies the implicit equation*

$$-\frac{2}{3r^2}\ln|\theta_1(t) - r| + \frac{1}{3r^2}\ln\left|\theta_1^2(t) + r\theta_1(t) + r^2\right| + \frac{2\,\text{sign}(r)}{r^2\sqrt{3}}\arctan\left(\frac{2\theta_1(t) + r}{|r|\sqrt{3}}\right) + C_0 = t \tag{48}$$

*and*

$$C_0 = \frac{2}{3r^2}\ln|\theta_0 - r| - \frac{1}{3r^2}\ln\left|\theta_0^2 + r\theta_0 + r^2\right| - \frac{2\,\text{sign}(r)}{r^2\sqrt{3}}\arctan\left(\frac{2\theta_0 + r}{|r|\sqrt{3}}\right). \tag{49}$$

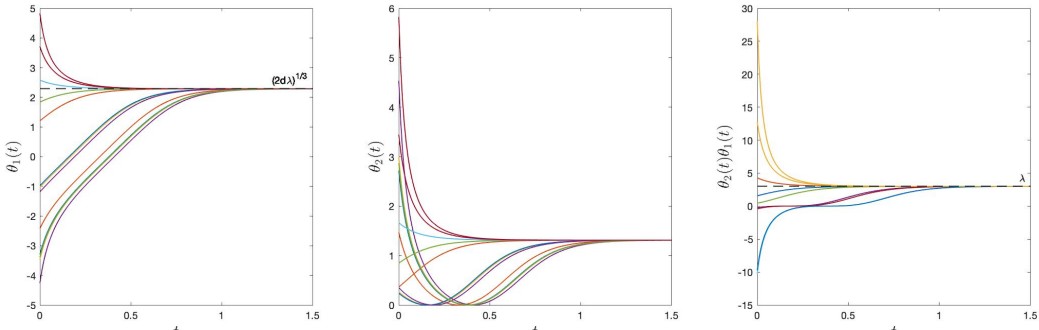

Figure 3: The plots of $\theta_1(t)$, $\theta_2(t)$ and the product $\theta_2(t)\theta_1(t)$ with initial condition $\theta_0 \sim \mathcal{U}(-5,5)$, $\lambda = 3$ and FA constant $d = 2$ (10 simulations).

To conclude the proof, we need to show that the product $\theta_2(t)\theta_1(t)$ is monotonic increasing (if $\theta_0 < r$) or decreasing (if $\theta_0 > r$) to the limit value $\lambda$. See Figure 3 for a comparative plot of the solutions $\theta_1(t)$, $\theta_2(t)$ and their product.

The monotonicity properties of $\theta_1$ are described in Lemma 12). We recall here the first derivative of the functions $\theta_1$ and $\theta_2 = \frac{1}{2d}\theta_1(t)^2$

$$\dot{\theta}_1 = \frac{1}{2}\left(r^3 - \theta_1^3\right), \qquad \dot{\theta}_2 = \frac{1}{d}\theta_1\dot{\theta}_1, \tag{50}$$

and its the second derivative

$$\ddot{\theta}_1 = \frac{3}{4}\theta_1^2\left(\theta_1^3 - r^3\right), \qquad \ddot{\theta}_2 = \frac{1}{d}\left((\dot{\theta}_1)^2 + \theta_1\ddot{\theta}_1\right) \tag{51}$$

If $\theta_0 > r$, then $\theta_1(t)$ is strictly convex and strictly decreasing; by construction, $\theta_2(t)$ is also strictly convex and decreasing. Therefore, the product $\theta_2(t)\theta_1(t)$ will be strictly convex and strictly decreasing $\forall\, t \in \mathbb{R}_{\geq 0}$.

If $\theta_0 < r$, then $\theta_1(t)$ is strictly increasing and by construction the product $\theta_2(t)\theta_1(t) = \frac{1}{2d}\theta_1^3$ is increasing $\forall\, t \in \mathbb{R}_{\geq 0}$. The description of convexity/concavity in this case is more involved and we do not pursue this direction, as it is not necessary for our results. □

**Case $\lambda = 0$.** If we assume the signal $\lambda = 0$, we can follow the same argument as above and perform similar calculations.

Equation (40) has a simple explicit solution (assuming $\theta_0 \neq 0$)

$$\theta_1(t) = \pm\sqrt{\frac{1}{t + \theta_0^{-2}}}, \qquad \theta_2(t) = \frac{1}{2d(t + \theta_0^{-2})}, \tag{52}$$

where $\pm$ depends on the sign of $\theta_0$. It is easy to see that

$$\theta_2(t)\theta_1(t) \to 0 = \lambda \qquad \text{as } t \to +\infty, \tag{53}$$

however the convergence rate is of the order of $\mathcal{O}(t^{-\frac{3}{2}})$, therefore not exponential.

### A.3 PROOF OF THEOREM 3

In the setting of Theorem 3, we are considering the initial conditions of $\theta_1$ and $\theta_2$ as independent, therefore the integration constant

$$K = \theta_2(0) - \frac{1}{2d}\theta_1(0)^2 \in \mathbb{R}$$

may assume any value. The cubic polynomial in the ODE for $\theta_1$ has the general form

$$\dot{\theta}_1 = -\frac{1}{2} \left( \theta_1^3 + 2dK\theta_1 - 2d\lambda \right) \tag{54}$$

and it could have 1 real root (and 2 complex conjugate roots), 2 real roots (a simple and a double root) or 3 real distinct roots. To discern the cases, we need to evaluate the discriminant:

$$\Delta = -4d^2 \left( 8dK^3 + 27\lambda^2 \right). \tag{55}$$

Regardless of the sign of the discriminant, there exists a unique global solution $\theta_1 \in C^1([0, +\infty))$ for any initial condition $\theta_1(0) \in \mathbb{R}$ and any constant $K \in \mathbb{R}$ (Picard-Lindelöf Theorem).

**Case $\Delta > 0$.** The polynomial has 3 real, distinct root $r_1 < r_2 < r_3$, that may be written explicitly with the help of trigonometric functions, if needed. Note also that $\Delta > 0$ implies $K < 0$. There are three constant solutions of the equation (54): $\theta_1(t) = r_j, \forall t \geq 0$ (provided that $\theta_0 = r_j$ for some $j = 1, 2, 3$). Otherwise ($\theta_0 \neq r_j \ \forall \ j = 1, 2, 3$), the implicit solution of (54) is the following:

$$\frac{\ln |\theta_1(t) - r_3|}{(r_3 - r_2)(r_3 - r_1)} - \frac{\ln |\theta_1(t) - r_2|}{(r_3 - r_2)(r_2 - r_1)} + \frac{\ln |\theta_1(t) - r_1|}{(r_3 - r_1)(r_2 - r_1)} + C_0 = -\frac{t}{2} \tag{56}$$

with constant of integration

$$C_0 = -\frac{\ln |\theta_0 - r_3|}{(r_3 - r_2)(r_3 - r_1)} + \frac{\ln |\theta_0 - r_2|}{(r_3 - r_2)(r_2 - r_1)} - \frac{\ln |\theta_0 - r_1|}{(r_3 - r_1)(r_2 - r_1)}. \tag{57}$$

**Lemma 14.** $\theta_1(t)$ *is monotonic:*

$$\dot{\theta}_1 < 0 \qquad \qquad if \quad \theta_0 \in (r_1, r_2) \cup (r_3, +\infty), \tag{58}$$

$$\dot{\theta}_1 > 0 \qquad \qquad if \quad \theta_0 \in (-\infty, r_1) \cup (r_2, r_3). \tag{59}$$

In particular, from Lemma 14 we can conclude that the constant solutions $\theta_1(t) = r_1$ and $\theta_1(t) = r_3$ are attractive and $\theta_1(t) = r_2$ is repulsive; the general solution (regardless of the initial value $\theta_0 \in \mathbb{R}$) is bounded for all times.

Following similar arguments as in Section A.2, we can conclude that the convergence is exponentially fast:

$$|\theta_1(t) - r_3| = e^{-\frac{(r_3 - r_2)(r_3 - r_1)}{2}t + \mathcal{O}(1)} \qquad \text{as } t \to +\infty, \tag{60}$$

for $\theta_0 \in (r_2, r_3) \cup (r_3, +\infty)$. A similar asymptotic expansion is valid for the quantity $|\theta_1(t) - r_1|$ when $\theta_0 \in (-\infty, r_1) \cup (r_1, r_2)$.

Using the relation $\theta_2(t) = \frac{1}{2d}\theta_1(t)^2 + K$, we have

$$\left| \theta_2(t)\theta_1(t) - \frac{1}{2d}r_3^3 - Kr_3 \right| \leq Ce^{-\frac{(r_3 - r_2)(r_3 - r_1)}{2}t} \tag{61}$$

for some $C > 0$. Notice that, being $r_3$ a root of the cubic polynomial $x^3 + 2dx - 2d\lambda$, it follows that $\frac{1}{2d}r_3^3 - Kr_3 = \lambda$. Similar arguments hold for $r_1$.

**Case $\Delta < 0$.** The polynomial has 1 real root $r \in \mathbb{R}$ and two complex conjugate roots. Let us assume $\lambda > 0$ (therefore, $r \neq 0$); the case $\lambda = 0$ is analyzed shortly below.

Using Cardano's formula, the single root has the explicit expression

$$r = \sqrt[3]{d\lambda + \sqrt{d^2\lambda^2 + \frac{8d^3K^3}{27}}} + \sqrt[3]{d\lambda - \sqrt{d^2\lambda^2 + \frac{8d^3K^3}{27}}}. \tag{62}$$

The setting is the similar to the one analyzed Section A.2. From a monotonicity argument, it follows that the constant solution $\theta_1(t) = r$ (with $\theta_0 = r$) is attractive, therefore for any given initial condition $\theta_0$, $\theta_1(t)$ is bounded for all times $t$ and it converges to $r$ as $t \to +\infty$.

The implicit solution reads

$$\frac{r}{2\left(r^3 + d\lambda\right)} \ln|\theta_1(t) - r| - \frac{r}{4\left(r^3 + d\lambda\right)} \ln\left(\theta_1(t)^2 + r\theta_1(t) + \frac{2d\lambda}{r}\right)$$
$$-\frac{3r^3}{8(r^3 + d\lambda)(d\lambda - r^3)} \arctan\left(\frac{r\left(2\theta_1(t) + r\right)}{4\left(d\lambda - r^3\right)}\right) + C_0 = -\frac{t}{2} \tag{63}$$

with $C_0$ the constant of integration.

From $\theta_2(t) = \frac{1}{2d}\theta_1(t)^2 + K$, convergence follows: $\theta_2(t)\theta_1(t) \to \frac{1}{2d}r^3 + Kr = \lambda$ as $t \to +\infty$. Furthermore, convergence is exponentially fast :

$$|\theta_2(t)\theta_1(t) - \lambda| \le Ce^{-\frac{r^3 + d\lambda}{r}t} \qquad \text{as } t \to +\infty, \tag{64}$$

for some $C > 0$.

**Case $\Delta = 0$.** The vanishing of the discriminant implies that the constant of integration $K$ has a specific value

$$K = -\frac{3}{2}\sqrt[3]{\frac{\lambda^2}{d}}.$$

**Remark 15.** *Notice that the set of initial conditions $(\theta_1(0), \theta_2(0))$ such that $\Delta = 0$ is a set of Lebesgue measure zero in $\mathbb{R}^2$.*

In this case, the polynomial has 1 simple root $r_s$ and one double root $r_d$:

$$r_s = -\frac{3\lambda}{K} = 2\sqrt[3]{d\lambda}, \qquad r_d = \frac{3\lambda}{2K} = -\sqrt[3]{d\lambda} \tag{65}$$

The solution of the differential equation (54) (for $\theta_0 \ne r_s, r_d$) reads

$$\frac{\ln|\theta_1(t) - r_s| - \ln|\theta_1(t) - r_d|}{(r_s - r_d)^2} + \frac{1}{(r_s - r_d)(\theta_1(t) - r_d)} + C_0 = -\frac{t}{2} \tag{66}$$

with $C_0$ the constant of integration.

**Lemma 16.** *The solution $\theta_1(t)$ is monotonic: if $r_s > r_d$,*

$$\dot\theta_1 < 0 \qquad\qquad\qquad\qquad \text{if} \quad \theta_0 \in (r_s, +\infty) \tag{67}$$
$$\dot\theta_1 > 0 \qquad\qquad \text{if} \quad \theta_0 \in (-\infty, r_d) \cup (r_d, r_s) \tag{68}$$

*and similarly if $r_s < r_d$,*

$$\dot\theta_1 < 0 \qquad\qquad \text{if} \quad \theta_0 \in (r_s, r_d) \cup (r_d, +\infty) \tag{69}$$
$$\dot\theta_1 > 0 \qquad\qquad\qquad\qquad \text{if} \quad \theta_0 \in (-\infty, r_s). \tag{70}$$

Therefore, the solution $\theta_1(t) = r_s$ is always attractive, while the solution $\theta_1(t) = r_d$ is "one-sided" attractive. The solution $\theta_1(t)$ is bounded for all times and the convergence result follows:

$$\theta_2(t)\theta_1(t) \to \frac{1}{2d}r_s^3 + Kr_s = \lambda \qquad \text{as } t \to \infty; \tag{71}$$

similarly, for $r_d$. Exponential convergence holds when $\theta_1(t) \to r_s$, while when $\theta_1(t) \to r_d$ the rate is sub-exponential.

In the special case when $\Delta = 0$ and $\lambda = 0$, the polynomial has a triple root $r = 0$. The solution is the same as the one derived in Section A.2 in the case of $\lambda = 0$ and the convergence to the signal $\lambda = 0$ is polynomial:

$$|\theta_2(t)\theta_1(t)| \le Ct^{-\frac{3}{2}} \qquad \text{as } t \to +\infty, \tag{72}$$

for some $C > 0$.

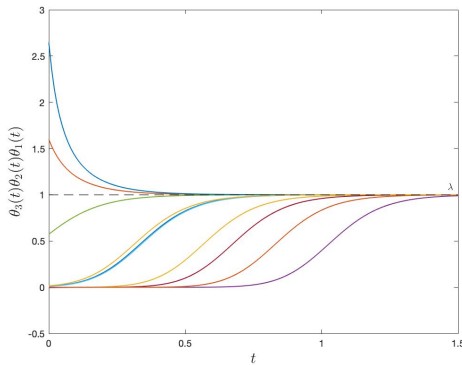

Figure 4: The plot of the product $\theta_3(t)\theta_2(t)\theta_1(t)$ with initial conditions $\theta_0 \sim \mathcal{U}([-1,2])$, FA constants $d_1 = 2$ and $d_2 = 2.5$, $\lambda = 1$ (10 simulations).

## B  CONVERGENCE OF FA FOR $L$-LAYER NNS

### B.1  PROOF OF PROPOSITION 4

We have that $\mathcal{L} := \frac{1}{2}\mathbb{E}[\|\boldsymbol{y} - \hat{\boldsymbol{y}}\|^2]$ and $\hat{\boldsymbol{y}} = \boldsymbol{W}_L \cdots \boldsymbol{W}_2 \boldsymbol{W}_1 \boldsymbol{x}$, thus noting $\boldsymbol{W}_{[a:b]} := \boldsymbol{W}_b \boldsymbol{W}_{b-1} \cdots \boldsymbol{W}_a$ we have,

$$\nabla_{W_\ell}\mathcal{L} = \mathbb{E}[\tfrac{1}{2}\nabla_{W_\ell}\|\boldsymbol{y} - \hat{\boldsymbol{y}}\|^2] \tag{73}$$

$$= \mathbb{E}[\boldsymbol{W}_{[\ell+1:L]}^\top(\boldsymbol{y} - \boldsymbol{W}_L \cdots \boldsymbol{W}_1 \boldsymbol{x})\boldsymbol{x}^\top \boldsymbol{W}_{[1:i-1]}^\top] \tag{74}$$

$$= \boldsymbol{W}_{[\ell+1:L]}^\top(\boldsymbol{\Sigma}_{xy} - \boldsymbol{W}_L \cdots \boldsymbol{W}_1 \boldsymbol{\Sigma}_{xx})\boldsymbol{W}_{[1:\ell-1]}^\top, \tag{75}$$

where the last line comes from the linearity of the expectation. Now we conclude by noticing that by definition of the FA dynamics (2), the matrices on the left hand-side of $(\boldsymbol{y} - \boldsymbol{W}_2 \boldsymbol{W}_1 \boldsymbol{x})$ are replaced by random matrices $\tilde{\boldsymbol{M}}_l$. Thus we get

$$\dot{\boldsymbol{W}}_\ell = \boldsymbol{M}_\ell(\boldsymbol{\Sigma}_{xy} - \boldsymbol{W}_{[1:L]}\boldsymbol{\Sigma}_{xx})\boldsymbol{W}_{[1:\ell-1]}^\top \qquad \ell \in [L], \tag{76}$$

where we reparametrized $\boldsymbol{M}_\ell = \boldsymbol{M}_{[\ell+1:L]}$ (see discussion in the main text).

### B.2  PROOF OF THEOREM 5

We will first state Theorem 5 in a more extensive form. Given the system

$$\dot{\theta}_L = (\lambda - \theta_L \ldots \theta_1)\theta_{L-1} \ldots \theta_1 \tag{77}$$

$$\dot{\theta}_{L-1} = d_{L-1}(\lambda - \theta_L \ldots \theta_1)\theta_{L-2} \ldots \theta_1 \tag{78}$$

$$\vdots \tag{79}$$

$$\dot{\theta}_2 = d_2(\lambda - \theta_L \ldots \theta_1)\theta_1 \tag{80}$$

$$\dot{\theta}_1 = d_1(\lambda - \theta_L \ldots \theta_1) \tag{81}$$

where we suppressed the index $i$ for simplicity, the following result holds.

**Theorem 17.** *For any set of FA constants $\{d_\ell\}_{\ell=1,\ldots,L-1}$, $d_\ell \in \mathbb{R}_{>0}$, there exists an initialization scheme such that the system of differential equations (77)–(81) can be reduced into an equation for $\theta_1$ of the form*

$$\dot{\theta}_1 = d_1\left(\lambda - \mathfrak{K}\theta_1^\gamma\right) \tag{82}$$

*for some constant $\mathfrak{K} \in \mathbb{R}_{>0}$ that depends on $\{d_\ell\}$, with $\gamma = 2^L - 1$. The other functions depend directly on $\theta_1$ in a power-like fashion:*

$$\theta_\ell(t) = \frac{C_\ell}{2^{\ell-1}}\theta_1(t)^{2^{\ell-1}}, \qquad \ell = 2, \ldots, L, \tag{83}$$

*for some $C_\ell \in \mathbb{R}_{>0}$. Furthermore, the following convergence result holds:*

$$\theta_L(t)\theta_{L-1}(t)\dots\theta_1(t) \to \lambda \qquad as \ t \to +\infty. \tag{84}$$

See Figure 4 for a numerical simulation of a 3-layer scalar network.

*Proof.* The proof of reducing the system of differential equations into a single equation for $\theta_1$ will be constructive, following a simple iterative procedure. We start by considering the last equation (81) and by substituting $(\lambda - \theta_L \dots \theta_1) = \frac{\dot{\theta}_1}{d_1}$ in the second to last equation

$$\dot{\theta}_2 = \frac{d_2}{d_1}\theta_1\dot{\theta}_1 = \frac{d_2}{2d_1}\left(\theta_1^2\right)^{\cdot}, \qquad \text{implying} \quad \theta_2(t) = \frac{d_2}{2d_1}\theta_1(t)^2 + K_2, \tag{85}$$

for some constant of integration $K_2 \in \mathbb{R}$, which can be set to zero if we impose $\theta_2(0) = \frac{d_2}{d_1}\theta_1(0)^2$.

The same substitution strategy can be consecutively applied to the third to last equation, fourth to last equation and so on. In general, it is easy to prove by induction that $\forall \ell = 2, \dots, L$ we have $\dot{\theta}_\ell = C_\ell \theta_1^{2^{\ell-1}-1}\dot{\theta}_1$, for some $C_\ell = C_\ell(d_1, \dots, d_\ell) \in \mathbb{R}_{>0}$, which implies

$$\theta_\ell(t) = \frac{C_\ell}{2^{\ell-1}}\theta_1(t)^{2^{\ell-1}} + K_\ell \tag{86}$$

where as before we can set the constant of integration $K_\ell = 0$ by imposing $\theta_\ell(0) = \frac{C_\ell}{2^{\ell-1}}\theta_1(0)^{2^{\ell-1}}$.

Once we have derived the relations between the functions $\{\theta_\ell(t)\}_{\ell=2,\dots,L}$ and $\theta_1(t)$, we can plug all of them into (81) to obtain a closed-form differential equation for $\theta_1$:

$$\dot{\theta}_1 = d_1\left(\lambda - \mathfrak{K}\theta_1^\gamma\right), \tag{87}$$

where $\mathfrak{K} = \prod_{\ell=1}^{L} \frac{C_\ell}{2^{\ell-1}} \in \mathbb{R}_{>0}$ and $\gamma = \sum_{\ell=0}^{L-1} 2^\ell = 2^L - 1$. This is a separable equation which can be solved (with tears) by using partial fraction decomposition.

The proof of convergence of the product $\prod_{\ell=1}^{L}\theta_\ell(t)$ to the true signal $\lambda$ follows a similar argument as in Appendix A.2. Let $r := \sqrt[\gamma]{\frac{\lambda}{\mathfrak{K}}}$ be the unique real root of the polynomial $\lambda - \mathfrak{K}\theta_1^\gamma$, then the solution $\theta_1(t) = r$ is a constant solution, provided that $\theta_1(0) = r$. By construction, it follows that $\{\theta_\ell(t)\}_{\ell=2,\dots,L}$ are also constant functions and their product is such that

$$\theta_L(t)\dots\theta_1(t) = \prod_{\ell=1}^{L}\frac{C_\ell}{2^{\ell-1}}r^\gamma = \lambda \qquad \forall t \geq 0. \tag{88}$$

**Lemma 18.** *The general solution $\theta_1(t)$ is monotonic:*

$$\dot{\theta}_1 > 0 \qquad\qquad\qquad\quad if \quad \theta_1(0) < r \tag{89}$$

$$\dot{\theta}_1 < 0 \qquad\qquad\qquad\quad if \quad \theta_1(0) > r \tag{90}$$

Therefore, $\theta_1(t)$ is bounded for all times and $\theta_1(t) \to r$ as $t \to +\infty$. By construction, it follows that

$$\theta_L(t)\dots\theta_1(t) \to \prod_{\ell}\frac{C_\ell}{2^{\ell-1}}r^\gamma = \lambda, \qquad as \ t \to +\infty. \tag{91}$$

Exponential convergence follows from a careful analysis of the implicit solution of the separable equation (87), in a similar way as it has been done in Appendix A.2. $\square$

The proof of convergence of the FA algorithm for a generic initialization $(\theta_1(0), \dots, \theta_L(0))$, where each layer is initialized independently, follows the same guidelines as the 2-layer case (Appendix A.3).

*Sketch of the proof.* In the same setting as the proof above, by keeping all the constants of integration $K_\ell \neq 0$, we reduce the system of ODEs (81) to a single ODE for $\theta_1$ of the form

$$\dot{\theta}_1 = \mathcal{P}_\gamma(\theta_1; d_1, \dots, d_L, \lambda) \tag{92}$$

where $\mathcal{P}$ is a polynomial of degree $\gamma := 2^L - 1$ in the variable $\theta_1$ with coefficients depending on the FA constants $d_1, \dots, d_L$ and the signal $\lambda$. It is still a separable equation that can be studied via a careful analysis of its discriminant. $\square$

## C  IMPLICIT REGULARIZATION PHENOMENA

### C.1  PROOF OF THEOREM 7 (CASE $\Delta > 0$)

For readability, we are suppressing the index $i$ from all the formulæ below. Recall the Initial Value Problem for $\theta_1$ (and $\theta_2$):

$$\dot{\theta}_1 = -\frac{1}{2}\left(\theta_1^3 + 2dK\theta_1 - 2d\lambda\right) \tag{93}$$

$$\theta_1(0) = \theta_0 \tag{94}$$

and $\theta_2(t) = \frac{1}{2d}\theta_1^2(t) + K$, where $K = \theta_2(0) - \frac{1}{2d}\theta_0^2$. By assuming that the discriminant is positive $\Delta = -4d^2\left(8dK^3 + 27\lambda^2\right) > 0$, the cubic polynomial on the right hand side of (93) has 3 real distinct root $r_1 < r_2 < r_3$ and the solution of the ODE is (see also Section A.3)

$$\frac{\ln|\theta_1(t) - r_3|}{r_{32}r_{31}} - \frac{\ln|\theta_1(t) - r_2|}{r_{32}r_{21}} + \frac{\ln|\theta_1(t) - r_1|}{r_{31}r_{21}} + C_0 = -\frac{t}{2} \tag{95}$$

$$C_0 = -\frac{\ln|\theta_0 - r_3|}{r_{32}r_{31}} + \frac{\ln|\theta_0 - r_2|}{r_{32}r_{21}} - \frac{\ln|\theta_0 - r_1|}{r_{31}r_{21}}, \tag{96}$$

where we set $r_{ij} := r_i - r_j$.

If we set as initial value $\theta_0 = r_2 + e^{-\delta}$ ($\delta > 0$), then

$$C_0 = -\frac{\delta}{r_{32}r_{21}} - \frac{\ln\left(r_{32} - e^{-\delta}\right)}{r_{32}r_{31}} - \frac{\ln\left(r_{21} + e^{-\delta}\right)}{r_{31}r_{21}}; \tag{97}$$

we additionally rescale the time variable as $t \mapsto \delta t$. Rewriting equation (95), we have

$$r_3 - \theta_1(\delta t) = \exp\left\{-\frac{r_{32}r_{31}}{2}\delta\left[t - \frac{2}{r_{32}r_{21}}\right] + \ln\left(r_{32} - e^{-\delta}\right) + \frac{r_{32}}{r_{21}}\ln\left(r_{21} + e^{-\delta}\right)\right.$$
$$\left. + \frac{r_{31}}{r_{21}}\ln\left(\theta_1(\delta t) - r_2\right) - \frac{r_{32}}{r_{21}}\ln\left(\theta_1(\delta t) - r_1\right)\right\}, \tag{98}$$

$$\theta_1(\delta t) - r_2 = \exp\left\{-\frac{r_{32}r_{21}}{2}\delta\left[\frac{2}{r_{32}r_{21}} - t\right] + \frac{r_{21}}{r_{31}}\ln\left(\frac{r_3 - \theta_1(\delta t)}{r_{32} - e^{-\delta}}\right) + \frac{r_{32}}{r_{31}}\ln\left(\frac{\theta_1(\delta t) - r_1}{r_{21} + e^{-\delta}}\right)\right\}. \tag{99}$$

From (98)–(99), we have the following asymptotic expansions: as $\delta \to +\infty$

$$\theta_1(\delta t) \to r_3 \qquad\qquad\qquad\qquad \text{for } t > T \tag{100}$$

$$\theta_1(\delta T) \to \alpha := r_3 - \exp\left\{\left(1 + \frac{r_{31}}{r_{21}}\right)\ln r_{32} + \frac{r_{32}}{r_{21}}\ln\left(\frac{r_{21}}{r_{31}}\right)\right\} \tag{101}$$

$$\theta_1(\delta t) \to r_2 \qquad\qquad\qquad\qquad \text{for } t < T \tag{102}$$

with $T = \frac{2}{r_{32}r_{21}}$.

In the case $\theta_0 = r_2 - e^{-\delta}$, we can follow a similar argument: as $\delta \to +\infty$,

$$\theta_1(\delta t) \to r_1 \qquad\qquad\qquad\qquad \text{for } t > T \tag{103}$$

$$\theta_1(\delta T) \to \tilde{\alpha} := r_1 + \exp\left\{\left(1 + \frac{r_{31}}{r_{32}}\right)\ln r_{21} + \frac{r_{21}}{r_{32}}\ln\left(\frac{r_{32}}{r_{31}}\right)\right\} \tag{104}$$

$$\theta_1(\delta t) \to r_2 \qquad\qquad\qquad\qquad \text{for } t < T. \tag{105}$$

The last bit of result we need is to analyze if the value of the critical value

$$T = \frac{2}{r_{32}r_{21}} \tag{106}$$

is monotonic as $\lambda$ increases. Recall the cubic polynomial on the right hand side of equation (93) (without loss of generality, we are setting $2d = 1$): $P(x) = x^3 + Kx - \lambda = (x - r_1)(x - r_2)(x - r_3)$, where $\lambda > 0$ and $K < -\frac{3}{2}\lambda^{2/3} < 0$ depends on the initial conditions of the FA system.

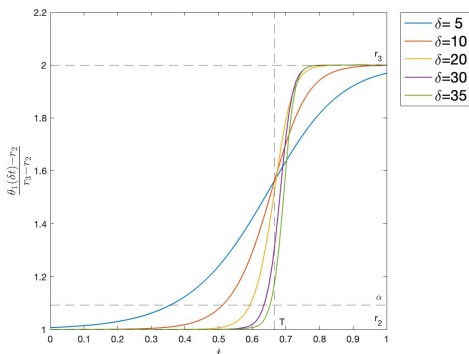 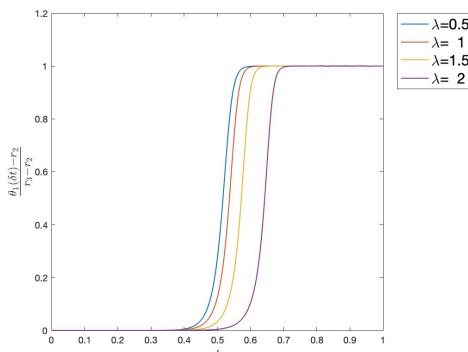

(a) Numerical simulation of $\theta_1(t)$ with $r_1 = -2$, $r_2 = 1$, $r_3 = 2$ and initial conditions $\theta_0 = r_2 + e^{-\delta}$ with $\delta \in \{10, 20, 30\}$.

(b) Plot of $\theta_1(\delta t)$ ($\Delta > 0$), with initial condition $\theta_0 = r_2 + e^{-\delta}$, $\delta = 20$), FA constant $d = 0.5$, initial condition $K = -4$.

Figure 5: (Anti)-implicit regularization.

The quantity $r_{32}r_{21}$ can be expressed as $r_{32}r_{21} = -P'(r_2) = -3r_2^2 - K$, where $r_2 < 0$ for $\lambda > 0$. It is easy to see that if $\lambda < \tilde{\lambda}$, then $|r_2| < |\tilde{r}_2|$ and consequently $r_{32}r_{21} = -r_2^2 - K > -\tilde{r}_2^2 - K = \widetilde{r_{32}r_{21}}$, i.e.

$$T_\lambda = \frac{2}{r_{32}r_{21}} < \frac{2}{\widetilde{r_{32}r_{21}}} = T_{\tilde{\lambda}} \tag{107}$$

Therefore the threshold time $T$ is an increasing function of $\lambda$. See Figure 5b: in the $\delta$-scaling limit we see indeed that for smaller values of $\lambda$, the step appears sooner.

## C.2 Evidence of incremental learning in the case $K = 0$

Consider again the system (93) in the case $K = 0$:

$$\dot{\theta}_1 = -\frac{1}{2}(\theta_1^3 - 2d\lambda) \tag{108}$$

$$\theta_1(0) = \theta_0 \tag{109}$$

and $\theta_2(t) = \frac{1}{2d}\theta_1^2(t)$. In this case, the solution reads (see also Section A.2)

$$-\frac{2}{3r^2} \ln|\theta_1(t) - r| + \frac{1}{3r^2} \ln\left(\theta_1^2(t) + r\theta_1(t) + r^2\right) + \frac{2}{r^2\sqrt{3}} \arctan\left(\frac{2\theta_1(t) + r}{r\sqrt{3}}\right) + C_0 = t \tag{110}$$

$$C_0 = \frac{2}{3r^2} \ln|\theta_0 - r| - \frac{1}{3r^2} \ln\left(\theta_0^2 + r\theta_0 + r^2\right) - \frac{2}{r^2\sqrt{3}} \arctan\left(\frac{2\theta_0 + r}{r\sqrt{3}}\right) \tag{111}$$

with $r := \sqrt[3]{2d\lambda}$.

As the limit value of the function $\theta_1(t)$ is the positive root $r$, $\theta_1(T_0) = 0$ (for some $T_0 > 0$) if and only if $\theta_0 < 0$. The value of $T_0$ can be easily derived from the implicit expression (110):

$$T_0 = \frac{\pi}{r^2 3\sqrt{3}} + \frac{1}{3r^2} \ln\left(\frac{(r - \theta_0)^2}{\theta_0^2 + r\theta_0 + r^2}\right) - \frac{2}{r^2\sqrt{3}} \arctan\left(\frac{2\theta_0 + r}{r\sqrt{3}}\right) \tag{112}$$

and in the regime as $\theta_0 \to -\infty$

$$T_0 = \frac{4\pi}{3r^2\sqrt{3}} - \frac{1}{\theta_0^2} + \mathcal{O}\left(\frac{1}{\theta_0^3}\right), \qquad \theta_0 \to -\infty. \tag{113}$$

It is now clear that for bigger $\lambda$'s (i.e. dominant singular values) the crossing of the $x$-axis happens sooner: for $\lambda > \tilde{\lambda}$ and fixed FA constant $d > 0$,

$$T_{0,\lambda} < T_{0,\tilde{\lambda}}. \tag{114}$$

# D CONVERGENCE OF THE DISCRETE ALGORITHMS

We will report here all the proofs of the convergence theorems for the discrete FA dynamics. We will first discuss convergence guarantees and convergence rates for the modified (mid-point) FA dynamics for both 2- and $L$-layer NNs ($L \geq 3$), as it is more straightforward. The convergence proof for the forward Euler discretization method can be found in Section D.2.

## D.1 CONVERGENCE OF FA UNDER THE "MIDPOINT" DISCRETIZATION SCHEME

We recall the 2-layer system that we want to analyze. For simplicity we are suppressing all the $i$ indices and we are denoting $(x_t, y_t)$ the unknown functions in the system:

$$x_{t+1} = x_t + \eta d \left( \lambda - x_t y_t \right) \tag{115}$$

$$y_{t+1} = y_t + \frac{\eta}{2} \left( \lambda - x_t y_t \right) \left( x_{t+1} + x_t \right) \tag{116}$$

We begin with proving a series of easy lemmas that will be instrumental for the proof of the theorem.

**Lemma 19.** *With initial condition $(x_0, y_0) = (0, 0)$, the second component of the solution of the system (115)–(116) reads*

$$y_t = \frac{x_t^2}{2d} \qquad \forall\, t \geq 0. \tag{117}$$

*Proof.* From equation (115), $\frac{x_{t+1} - x_t}{\eta d} = \lambda - x_t y_t$; substituting in equation (116), we obtain the result:

$$y_{t+1} = y_t + \frac{(x_{t+1} + x_t)(x_{t+1} - x_t)}{2d} = y_t + \frac{x_{t+1}^2 - x_t^2}{2d}$$

$$= y_0 + \sum_{j=0}^{t} \frac{x_{j+1}^2 - x_j^2}{2d} = \underbrace{y_0 - \frac{x_0^2}{2d}}_{=0} + \frac{x_{t+1}^2}{2d} \tag{118}$$

via a telescoping sum. $\qquad\square$

Substituting the expression of $y_t$ back in equation (115), we have a recurrence relation for $x_t$ of the form:

$$x_{t+1} = x_t + \eta d \lambda \left( 1 - \frac{x_t^3}{r^3} \right) =: f(x_t). \tag{119}$$

where $r := \sqrt[3]{2d\lambda}$.

**Lemma 20.** *With parameter*

$$\eta < \frac{2}{3r^2} = \frac{2}{3(2d\lambda)^{\frac{2}{3}}}, \tag{120}$$

*the function $f(x) = x + \eta d \lambda \left( 1 - \frac{x^3}{r^3} \right)$ maps the interval $[0, r]$ into itself: $\forall\, x \in [0, r]$, $f(x) \in [0, r]$.*

*Proof.* By inspecting the function $f$ and its derivative we can conclude that $f(x) \geq 0$, $\forall\, x \in [0, r]$ and $f$ is increasing on the interval $\left[0, \sqrt{\frac{2}{3\eta}}\right]$. Notice also that $f(r) = r$. Imposing (120) guarantees that $r < \sqrt{\frac{2}{3\eta}}$, therefore

$$\eta d \lambda = f(0) \leq f(x) \leq f(r), \qquad \forall\, x \in [0, r].$$

$\qquad\square$

**Lemma 21.** *In the same hypotheses as Lemmas 19 and 20, the sequence $\{x_t\}_{t\geq 0}$ is bounded, positive and increasing:*

$$0 \leq x_t \leq r \qquad and \qquad x_t \leq x_{t+1}, \qquad \forall\, t \geq 0. \tag{121}$$

*Proof.* For $t = 0$, trivially $0 \leq x_0 = 0 \leq r$. By induction, assume that $0 \leq x_t \leq r$, then by Lemma 20

$$0 \leq x_{t+1} = x_t + \eta d \left( \lambda - \frac{x_t^3}{2d} \right) = f(x_t) \leq f(r) = r. \tag{122}$$

Furthermore,

$$x_{t+1} = x_t + \frac{\eta}{2} \left( 2d\lambda - x_t^3 \right) \geq x_t + \frac{\eta}{2} \left( 2d\lambda - (\sqrt[3]{2d\lambda})^3 \right) = x_t. \tag{123}$$

$\square$

We can now prove Theorem 9 that we rewrite here for convenience:

**Theorem 22.** *With initial conditions* $(x_0, y_0) = (0, 0)$, $\forall \lambda, d > 0$ *and with step size* $\eta < \frac{2}{3(2d\lambda)^{\frac{2}{3}}}$, *the following convergence result hold:*

$$|x_t y_t - \lambda| \leq 3\lambda \left( 1 - \frac{\eta}{2}(2d\lambda)^{\frac{2}{3}} \right)^t. \tag{124}$$

**Remark 23.** *Note that if* $\eta = \frac{2}{(2d\lambda)^{\frac{2}{3}}}$ *(which is outside the range prescribed in Theorem 9) and* $x_0 = 0$, *we achieve convergence in one step:* $x_t = \sqrt[3]{2d\lambda} \; \forall \, t > 0$.

*Proof.* By Lemma 21, $\exists \, \ell \in \mathbb{R}_{\geq 0}$ such that $x_t \to \ell$ as $t \to +\infty$. Furthermore, by solving

$$\ell = \ell + \eta d\lambda \left( 1 - \frac{\ell^3}{r^3} \right) \tag{125}$$

it follows that $\ell = r = \sqrt[3]{2d\lambda}$. Finally, we have

$$|x_t - r| = \left| x_{t-1} - r + \frac{\eta}{2} \left( r^3 - x_{t-1}^3 \right) \right| = |x_{t-1} - r| \left| 1 - \frac{\eta}{2} \left( x_{t-1}^2 + r x_{t-1} + r^2 \right) \right|$$

$$\leq |x_{t-1} - r| \left( 1 - \frac{3\eta}{2} r^2 \right)$$

$$\leq r \left( 1 - \frac{3\eta}{2} r^2 \right)^t \tag{126}$$

Note that $0 < 1 - \frac{3\eta}{2} r^2 < 1$, since $\eta < \frac{2}{3r^2}$. In conclusion, thanks to the arguments above and Lemma 19, we have: as $t \to \infty$

$$|x_t y_t - \lambda| = \left| \frac{x_t^3}{2d} - \lambda \right| = \frac{1}{2d} \left| x_t^2 + r x_t + r^2 \right| |x_t - r|$$

$$\leq \frac{3r^2}{2d} |x_t - r| \leq \frac{3r^3}{2d} \left( 1 - \frac{3\eta}{2} r^2 \right)^t$$

$$= 3\lambda \left( 1 - \frac{3\eta}{2} r^2 \right)^t. \tag{127}$$

$\square$

For the general $L$-layer case, we first recall Theorem 10.

**Theorem 24.** *Consider the following system:*

$$\theta_L^{(t+1)} = \theta_L^{(t)} + \eta \theta_{L-1}^{\left(t+\frac{1}{2}\right)} \dots \theta_1^{\left(t+\frac{1}{2}\right)} \left( \lambda - \theta_L^{(t)} \dots \theta_1^{(t)} \right) \tag{128}$$

$$\theta_{L-1}^{(t+1)} = \theta_{L-1}^{(t)} + \eta d_{L-1} \theta_{L-2}^{\left(t+\frac{1}{2}\right)} \dots \theta_1^{\left(t+\frac{1}{2}\right)} \left( \lambda - \theta_L^{(t)} \dots \theta_1^{(t)} \right) \tag{129}$$

$$\vdots \tag{130}$$

$$\theta_2^{(t+1)} = \theta_2^{(t)} + \eta d_2 \theta_1^{\left(t+\frac{1}{2}\right)} \left( \lambda - \theta_L^{(t)} \dots \theta_1^{(t)} \right) \tag{131}$$

$$\theta_1^{(t+1)} = \theta_1^{(t)} + \eta d_1 \left( \lambda - \theta_L^{(t)} \dots \theta_1^{(t)} \right) \tag{132}$$

with zero initialization $\theta_\ell^{(0)} = 0$, $\forall \ell = 1, \dots, L$. Assume that $\lambda, d_\ell > 0$, $\forall \ell = 1, \dots, L-1$, and the constant step size satisfies

$$\eta < \frac{1}{d_1 \gamma \left(\mathfrak{K}\lambda^{\gamma-1}\right)^{\frac{1}{\gamma}}}$$

where $\gamma = 2^L - 1$ and $\mathfrak{K}$ is a suitable positive constant depending on the FA constants $d_1, \dots, d_{L-1}$. Then, the product $\prod_{\ell=1}^{L} \theta_\ell^{(t)}$ linearly converges to the true signal $\lambda$:

$$\left| \theta_L^{(t)} \dots \theta_1^{(t)} - \lambda \right| \leq C_\lambda \left( 1 - \eta d_1 \gamma \left(\mathfrak{K}\lambda^{\gamma-1}\right)^{\frac{1}{\gamma}} \right)^t \tag{133}$$

for some constant $C_\lambda \in \mathbb{R}_{>0}$ only dependent on $\lambda$.

*Proof.* We consider the last equation (132) and substitute

$$\lambda - \theta_L^{(t)} \dots \theta_1^{(t)} = \frac{\theta_1^{(t+1)} - \theta_1^{(t)}}{\eta d_1} \tag{134}$$

in the second to last one to obtain

$$\theta_2^{(t+1)} = \theta_2^{(t)} + \frac{d_2}{2d_1} \left[ \left(\theta_1^{(t+1)}\right)^2 - \left(\theta_1^{(t)}\right)^2 \right] \quad \Rightarrow \quad \theta_2^{(t)} = \frac{d_2}{2d_1} \left(\theta_1^{(t)}\right)^2 \tag{135}$$

by telescoping sum. The same substitution strategy can be consecutively applied to the third to last equation, fourth to last equation and so on. In general, it is easy to prove by induction that $\forall \ell = 2, \dots, L$ we have

$$\theta_\ell^{(t+1)} = \theta_\ell^{(t)} + C_\ell \left[ \left(\theta_1^{(t+1)}\right)^{2^{\ell-1}} - \left(\theta_1^{(t)}\right)^{2^{\ell-1}} \right] \quad \Rightarrow \quad \theta_\ell^{(t)} = C_\ell \left(\theta_1^{(t)}\right)^{2^{\ell-1}} \tag{136}$$

for some $C_\ell = C_\ell(d_1, \dots, d_\ell) \in \mathbb{R}_{>0}$.

Once derived the relations between the sequences $\{\theta_\ell^{(t)}\}_{\ell=2,\dots,L}$ and $\{\theta_1^{(t)}\}$, we can go back to the recurrence equation (132) for $\theta_1^{(t)}$ to obtain

$$\theta_1^{(t+1)} = \theta_1^{(t)} + \eta d_1 \left[ \lambda - \mathfrak{K} \left(\theta_1^{(t)}\right)^\gamma \right] \tag{137}$$

where $\mathfrak{K} = \prod_{\ell=1}^{L} C_\ell \in \mathbb{R}_{>0}$ and $\gamma = \sum_{\ell=0}^{L-1} 2^\ell = 2^L - 1$.

**Lemma 25.** *With initial condition $\theta_1^{(t)} = 0$ and for $\eta < \frac{1}{d_1 \gamma (\mathfrak{K}\lambda^{\gamma-1})^{\frac{1}{\gamma}}}$, the sequence $\{\theta_1^{(t)}\}$ is positive, increasing and bounded by $r := \sqrt[\gamma]{\frac{\lambda}{\mathfrak{K}}}$.*

*Proof.* The proof follows the same argument as in the 2-layer case. The odd polynomial $f(x) = x + \eta d_1 (\lambda - \mathfrak{K}x^\gamma)$ has at least one fixed points at $x = r$ and $f(0) = \eta d_1 \lambda > 0$. Furthermore, $f([0,r]) \subseteq [0,r]$. Indeed, the function is increasing on the interval $[0, x_{\max}]$ with $x_{\max} = \left( \frac{1}{\eta d_1 \gamma \mathfrak{K}} \right)^{\frac{1}{\gamma-1}}$. If $\eta < \frac{1}{d_1 \gamma (\mathfrak{K}\lambda^{\gamma-1})^{\frac{1}{\gamma}}}$ then $r < x_{\max}$, implying that

$$0 < \eta d_1 \lambda = f(0) \leq f(x) \leq f(r) = r, \qquad \forall x \in [0,r].$$

It follows (by induction) that $0 \leq \theta_1^{(t)} \leq r$ $\forall t \geq 0$ and $\theta_1^{(t)} \leq \theta_1^{(t+1)}$. $\qquad\square$

From Lemma 25 we can conclude that $\theta_1^{(t)} \to r$ and

$$\left| \theta_1^{(t)} - r \right| = \left| \theta_1^{(t-1)} - r - \eta d_1 \mathfrak{K} \left( \left(\theta_1^{(t-1)}\right)^\gamma - r^\gamma \right) \right| = \left| \theta_1^{(t-1)} - r \right| \left| 1 - \eta d_1 \mathfrak{K} \sum_{j=0}^{\gamma-1} \left(\theta_1^{(t-1)}\right)^{\gamma-1-j} r^j \right|$$

$$\leq \left| \theta_1^{(t-1)} - r \right| \left| 1 - \eta d_1 \mathfrak{K} \gamma r^{\gamma-1} \right| = \left| \theta_1^{(t-1)} - r \right| \left( 1 - \eta d_1 \mathfrak{K} \gamma \left( \frac{\lambda}{\mathfrak{K}} \right)^{\frac{\gamma-1}{\gamma}} \right)$$

$$\leq r \left( 1 - \eta d_1 \gamma \left(\mathfrak{K}\lambda^{\gamma-1}\right)^{\frac{1}{\gamma}} \right)^t. \tag{138}$$

Finally, we have the following linear rate of convergence

$$\left| \theta_L^{(t)} \dots \theta_1^{(t)} - \lambda \right| = \left| \mathfrak{K} \left( \theta_1^{(t)} \right)^{\gamma} - \lambda \right| \le C_\lambda \left| \theta_1^{(t)} - r \right| \le \tilde{C}_\lambda \left( 1 - \eta d_1 \gamma \left( \mathfrak{K} \lambda^{\gamma - 1} \right)^{\frac{1}{\gamma}} \right)^t. \tag{139}$$

$\square$

## D.2 DISCRETIZATION UNDER THE EULER FORWARD METHOD

By discretizing the FA dynamics via the standard Euler forward method, we obtain the following recurrence relation for the iterates:

$$x_{t+1} = x_t + \eta d \left( \lambda - x_t y_t \right) \tag{140}$$
$$y_{t+1} = y_t + \eta x_t \left( \lambda - x_t y_t \right) \tag{141}$$

with initial condition $x_0 = y_0 = 0$. If we use the same substitution strategy as before ($\lambda - x_t y_t = \frac{x_{t+1} - x_t}{\eta d}$), we get

$$y_{t+1} = y_t + \eta x_t \frac{x_{t+1} - x_t}{\eta d} = y_t + \frac{x_t}{d} (x_{t+1} - x_t)$$

$$= y_t + \frac{1}{2d} \left[ (x_{t+1} + x_t) - (x_{x+1} - x_t) \right] (x_{t+1} - x_t) = y_t + \frac{1}{2d} \left( x_{t+1}^2 - x_t^2 \right) - \frac{1}{2d} (x_{t+1} - x_t)^2$$

$$= \underbrace{y_0 - \frac{x_0^2}{2d}}_{=0} + \frac{1}{2d} x_{t+1}^2 - \frac{1}{2d} \sum_{j=1}^{t} (x_j - x_{j-1})^2, \tag{142}$$

i.e.

$$y_t = \frac{1}{2d} x_t^2 - \frac{1}{2d} S_t \qquad \forall \, t \ge 0, \tag{143}$$

where $S_t := \sum_{i=1}^{t} (x_i - x_{i-1})^2$ and $S_0 = 0$. Notice that $0 \le S_t \le S_{t+1} \, \forall \, t \ge 0$, by definition.

Plugging (143) back into equation (140), we get a highly nonlinear recurrence relation where *all* the past terms of the sequence (up to time $t$) are involved in determining the subsequent term $x_{t+1}$:

$$x_{t+1} = x_t + \frac{\eta}{2} \left( 2d\lambda - x_t^3 + x_t S_t \right). \tag{144}$$

We will now provide a proof of Theorem 8. The proof is quite long and it is divided into four parts for readability. We will first rewrite the statement in the following way:

**Theorem 26.** *Consider the sequence $\{(x_t, y_t)\}$ defined recursively in (143)-(144) with zero initialization $x_0 = y_0 = 0$. For any $\lambda, d > 0$, set the constant step size as*

$$\eta < \min \left\{ \frac{2}{3(S^* + 1)^2}, \frac{2}{\max_{(x,s) \in \mathcal{R}} P(x, S)} \right\}, \tag{145}$$

*where $P(x, S) = 2d\lambda - x^3 + xS$, $S^*$ is the unique positive solution of $P(x, x) = 0$, and $\mathcal{R}$ is the following compact, convex set*

$$\mathcal{R} := \left\{ (x, S) \in \mathbb{R}_{\ge 0}^2 \mid S \le x, \ P(x, S) \ge 0 \right\}. \tag{146}$$

*Then, the product $x_t y_t$ linearly converges to the true signal $\lambda$:*

$$|x_t y_t - \lambda| \le C q^t. \tag{147}$$

*for some $C > 0$ and $0 < q < 1$.*

**Proof of convergence - part 1.** In order to prove convergence of the sequence $\{x_t\}$, we will first consider a more general sequence defined by the following recurrence relation

$$x_{t+1} = x_t + \frac{\eta}{2} \left( 2d\lambda - x_t^3 + x_t \alpha_t \right) \tag{148}$$

where $\{\alpha_t\}$ is an auxiliary sequence that we assume to be increasing, positive and bounded (therefore convergent: $\alpha_t \to \alpha_\infty$ for some $\alpha_\infty \in \mathbb{R}_{>0}$).

Consider the following fixed-point problem:

$$x = f(x; \alpha) \qquad \text{with } f(x; \alpha) = x + \frac{\eta}{2}\left(2d\lambda - x^3 + \alpha x\right) \tag{149}$$

with $x \geq 0$ and $\alpha \in [0, \alpha_\infty]$. Notice that $\forall \alpha \geq 0$, we have that $f(0; \alpha) = \eta d\lambda > 0$ and $f(x; \alpha) \to -\infty$ as $x \to +\infty$. Therefore, $\forall \alpha \geq 0$, there exists a unique fixed point $\ell_\alpha > 0$ for the function $f$ (uniqueness follows by directly inspecting the cubic polynomial $f(\cdot; \alpha)$ with $\alpha$ fixed).

From $\frac{\partial f}{\partial \alpha} = \frac{\eta}{2}x \geq 0$ for $x \geq 0$, it follows that $f(x; \alpha) \leq f(x; \alpha')\ \forall\, \alpha \leq \alpha'$. Furthermore, $\ell_\alpha \leq \ell_{\alpha'}$: indeed, $\forall\, \alpha \geq 0$, $\ell_\alpha$ is the unique zero of the cubic polynomial $P_\alpha(x) = -x^3 + \alpha x + 2d\lambda$ on the positive real line ($P_\alpha(0) = 2d\lambda > 0$, $P_\alpha'(0) = \alpha > 0$ and $P_\alpha(x) \to -\infty$ as $x \to -\infty$); its partial derivative $\frac{\partial P_\alpha}{\partial \alpha} = x \geq 0$, for $x \geq 0$, implies that as $\alpha$ increases the zero $\ell_\alpha$ shifts rightwards on $\mathbb{R}$.

On the other hand, from

$$\frac{\partial f}{\partial x} = 1 + \frac{\eta}{2}\alpha - \frac{3}{2}x^2 \geq 0 \tag{150}$$

we obtain that the function $f$ is increasing on the interval $\left[0, \sqrt{\frac{2}{3\eta} + \frac{\alpha}{3}}\right]$. In order to ensure monotonicity on the interval $[0, \ell_\alpha]$, we tune the parameter $\eta$ such that

$$\ell_\alpha \leq \sqrt{\frac{2}{3\eta} + \frac{\alpha}{3}}. \tag{151}$$

By taking the supremum over $\alpha \in [0, \alpha_\infty]$ on the left hand side and the infimum on the right hand side, we have

$$\ell_{\alpha_\infty} \leq \sqrt{\frac{2}{3\eta}}; \tag{152}$$

such a bound is satisfied if

$$\eta \leq \frac{2}{3\ell_{\alpha_\infty}^2}. \tag{153}$$

As it will be clear in the next parts of the proof (see Remark 29), the value of $\ell_{\alpha_\infty}$ is smaller than $S^*$, therefore (145) guarantees that the above bound (153) is satisfied. This implies that

1. $f(x; \alpha) \leq f(x'; \alpha)\ \forall\, x, x' \in [0, \ell_{\alpha_\infty}], x \leq x'$
2. $f(x; \alpha)$ maps $[0, \ell_{S_\infty}]$ into itself $\forall\, \alpha \in [0, \alpha_\infty]$.

In conclusion, we have

$$f(x; \alpha) \leq f(x'; \alpha') \qquad \forall\, x \leq x', \forall\, \alpha \leq \alpha'. \tag{154}$$

We are now ready to prove the following lemma.

**Lemma 27.** *For $\eta \leq \frac{2}{3\ell_{\alpha_\infty}^2}$, the sequence (148) (with zero initial condition) is positive, increasing and bounded: $\forall\, t \geq 0$*

$$0 \leq x_t \leq x_{t+1} \leq \ell_{\alpha_\infty} \tag{155}$$

*Proof.* The proof is by induction: for $t = 0$ we have $0 = x_0 \leq x_1 = \eta d\lambda \leq \ell_{\alpha_\infty}$ and assuming that $x_{t-1} \leq x_t$, it follows that

$$0 \leq x_t = f(x_{t-1}; \alpha_{t-1}) \leq f(x_t; \alpha_t) = x_{t+1} \leq f(\ell_{\alpha_\infty}; \alpha_\infty) = \ell_{\alpha_\infty} \tag{156}$$

$\square$

Therefore the sequence (148) converges: $x_t \to \omega$ for some $\omega \in \mathbb{R}_{>0}$. By construction, it is clear that $\omega = \ell_{\alpha_\infty}$. Indeed, given

$$x_{t+1} = x_t + \frac{\eta}{2}\left(2d\lambda - x_t^3 + x_t\alpha_t\right) \tag{157}$$

and taking the limit as $t \to \infty$ on both sides of the equation, we obtain

$$\omega = \omega + \frac{\eta}{2}\left(2d\lambda - \omega^3 + \omega\alpha_\infty\right), \tag{158}$$

i.e. $\omega$ is a (positive) solution of the fixed point problem $x = f(x; \alpha_\infty)$ whose unique solution is $\ell_{\alpha_\infty}$.

A major problem arises when we consider the case of $\alpha_t = S_t = \sum_{i=1}^t (x_i - x_{i-1})^2$, i.e. the auxiliary sequence $\{\alpha_t\}$ depends directly on the primary sequence $\{x_t\}$. Clearly, $0 \le \alpha_t \le \alpha_{t+1}$ for all times $t \ge 0$, but the boundedness property is not straightforward.

**Boundedness of the sequence of partial sums $\{S_t\}$.** Since $S_{t+1} = S_t + (x_{t+1} - x_{x_t})^2$, consider the recurrence relations:

$$x_{t+1} = x_t + \frac{\eta}{2}\left(2d\lambda - x_t^3 + x_t S_t\right) \tag{159}$$

$$S_{t+1} = S_t + \frac{\eta^2}{4}\left(2d\lambda - x_t^3 + x_t S_t\right)^2 \tag{160}$$

**Lemma 28.** *Given the convex, compact set*

$$\mathcal{R} := \left\{(x, S) \in \mathbb{R}_{\ge 0}^2 \mid S \le x,\ P(x, S) \ge 0\right\}, \tag{161}$$

*with* $P(x, S) = 2d\lambda - x^3 + xS$, *and*

$$\eta < \min\left\{\frac{2}{3(S^* + 1)^2}, \frac{2}{\max_{(x,s)\in\mathcal{R}} P(x, S)}\right\}, \tag{162}$$

*where $S^*$ is the unique positive solution to $P(x, x) = 0$, and with initial values $x_0 = S_0 = 0$, the sequence $\{(x_t, S_t)\}$ defined recursively in (159)-(160) is bounded and lies in the set $\mathcal{R}$.*

See Figure 6 for a sketch of the region $\mathcal{R}$.

*Proof.* The set of fixed points of the system is the set of zeros of the function $P(x, S)$,

$$\begin{cases} x = x + \frac{\eta}{2}P(x, S) \\ S = S + \frac{\eta^2}{4}P(x, S)^2 \end{cases} \qquad \Leftrightarrow \qquad P(x, S) = 2d\lambda - x^3 + xS = 0, \tag{163}$$

which is an algebraic curve in $\mathbb{R}^2$, with solutions in $(x, S) \in \mathbb{R}_{\ge 0}^2$

$$S = x^2 - \frac{2d\lambda}{x}; \tag{164}$$

furthermore, there exists a unique solution $(S^*, S^*) \in \mathbb{R}_{>0}^2$ such that $P(S^*, S^*) = 0$: indeed, $S^*$ is the unique solution to the equation $x^3 - x^2 + 2d\lambda = 0$ and it's explicit expression can be recovered from Cardano's formula (if needed).

We will prove the theorem by induction. For $t = 0$, trivially $0 = S_0 = x_0$ and $P(x_0, S_0) = 2d\lambda > 0$. For $t = 1$,

$$S_1 = \eta^2 d^2 \lambda^2 \le x_1 = \eta d\lambda \qquad \text{if } \eta \le \frac{1}{d\lambda} \tag{165}$$

$$P(x_1, S_1) = 2d\lambda - (\eta d\lambda)^3 + \eta d\lambda \cdot (\eta d\lambda^2) = 2d\lambda > 0. \tag{166}$$

It is easy to notice that if (162) is satisfied, then $\eta \le \frac{1}{d\lambda}$ as required in (165).

Since $\mathcal{R}$ is compact, the value of the polynomial $P(x, S)$ is bounded and positive for all $(x, S) \in \mathcal{R}$:

$$0 < 2d\lambda = P(x_0, S_0) \le \max_{(x,s)\in\mathcal{R}} P(x, S) < +\infty.$$

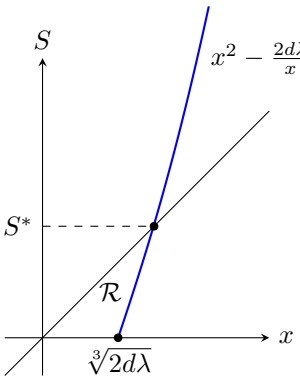

Figure 6: The compact, convex set $\mathcal{R} \subset \mathbb{R}_{\geq 0}^2$.

Assume that $\forall\, s = 1, \ldots, t$ we have

$$S_s \leq x_s, \qquad P(x_s, S_s) = 2d\lambda - x_s^3 + x_s S_s \geq 0 \tag{167}$$

then,

$$S_{t+1} = \underbrace{S_t}_{\leq x_t} + \frac{\eta^2}{4} P(x_t, S_t)^2 \leq x_t + \frac{\eta}{2} P(x_t, S_t) \cdot \underbrace{\frac{\eta}{2} P(x_t, S_t)}_{\leq 1} \leq x_t + \frac{\eta}{2} P(x_t, S_t) = x_{t+1},$$

$$\tag{168}$$

thanks to the induction step and (162). Additionally,

$$P(x_{t+1}, S_{t+1}) = P(x_t, S_t) + \nabla P(\bar{x}, \bar{S}) \cdot \begin{bmatrix} \frac{\eta}{2} P(x_t, S_t) \\ \frac{\eta^2}{4} P(x_t, S_t)^2 \end{bmatrix} \tag{169}$$

for some $(\bar{x}, \bar{S})$ belonging to the line segment connecting the point $(x_t, S_t)$ to the point $(x_{t+1}, S_{t+1})$, by the Mean Value Theorem;

$$
\begin{aligned}
P(x_{t+1}, S_{t+1}) &= P(x_t, S_t) + \frac{\eta}{2} P(x_t, S_t) \left(-3\bar{x} + \bar{S}\right) + \frac{\eta^2}{4} \bar{x} P(x_t, S_t)^2 \\
&= P(x_t, S_t) \left[1 + \frac{\eta}{2}(-3\bar{x}^2 + \bar{S}) + \frac{\eta^2}{4} \bar{x} P(x_t, S_t)\right] \\
&\geq P(x_t, S_t) \left[1 - \frac{3\eta}{2} \bar{x}^2\right].
\end{aligned}
\tag{170}
$$

Notice now that $0 \leq \bar{x} = x_t + \delta(x_{t+1} - x_t) = x_t + \delta \frac{\eta}{2} P(x_t, S_t)$, for some $\delta \in [0, 1]$; since $(x_t, S_t) \in \mathcal{R}$ by induction,

$$\bar{x} \leq S^* + 1 \cdot \underbrace{\frac{\eta}{2} \max_{(x,S) \in \mathcal{R}} P(x, S)}_{\leq 1} \leq S^* + 1. \tag{171}$$

Therefore,

$$P(x_{t+1}, S_{t+1}) \geq P(x_t, S_t) \left[1 - \frac{3\eta}{2} \bar{x}^2\right] \geq P(x_t, S_t) \left[1 - \frac{3\eta}{2} \left(S^* + 1\right)^2\right] \geq 0, \tag{172}$$

by (162). The result follows. $\qquad\square$

**Proof of convergence - part 2.** Theorem 28 above implies that the sequence of partial sums $\{S_t = \sum_{i=1}^t (x_i - x_{i-1})^2\}$ converges: $S_t \to S_\infty := \sum_{i=0}^\infty (x_i - x_{i-1})^2 < +\infty$ and we can resort to the arguments in the first part of the proof to conclude that $x_t \to \ell_{S_\infty}$.

**Remark 29.** *Note that $\ell_{S_\infty}$ is the unique positive solution to the equation $P(x, S_\infty) = 0$ and, since the sequence $\{(x_t, S_t)\} \subset \mathcal{R}$, $\ell_{S_\infty}$ lies on the boundary of $\mathcal{R}$. This implies that $\ell_{S_\infty} \leq S^*$.*

Finally, the product $x_t y_t \to \lambda$ as $t \to +\infty$. Indeed, since $x_t \to \ell_{S_\infty} =: \mathcal{L}$, then, by construction $y_t$ will also converge to some value $\hat{\mathcal{L}}$ (because of (143)). Moreover, $\mathcal{L}$ is the fixed point of the recurrence equation (144):

$$\mathcal{L} = \mathcal{L} + \frac{\eta}{2}\left(2d\lambda - \mathcal{L}^3 + \mathcal{L}S_\infty\right), \tag{173}$$

which can be alternatively written as

$$\mathcal{L} = \mathcal{L} + \eta d\left(\lambda - \mathcal{L}\hat{\mathcal{L}}\right); \tag{174}$$

$\mathcal{L}$ is fixed point if and only if $\mathcal{L}\hat{\mathcal{L}} = \lambda$, implying $x_t y_t \to \lambda$.

**Linear rates.** The last piece of results that needs to be proven is the linear rate of convergence. Consider the difference $\ell_{S_t} - x_t$, where $\ell_{S_t}$ is the unique positive solution to $P(x, S_t) = 0$ and by construction $x_t \leq S_t$:

$$\begin{aligned}
\ell_{S_{t+1}} - x_{t+1} - &= \ell_{S_{t+1}} - x_t - \frac{\eta}{2}P(x_t, S_t) \\
&= \ell_{S_{t+1}} - x_t - \frac{\eta}{2}\left(\ell_{S_t} - x_t\right)\left(x_t^2 + \ell_{S_t}x_t + \frac{2d\lambda}{\ell_{S_t}}\right) \\
&\leq \left(\ell_{S_t} - x_t\right)\underbrace{\left[1 - \frac{\eta}{2}\left(x_t^2 + x_t\ell_{S_t} + \frac{2d\lambda}{\ell_t}\right)\right]}_{=:B_1} + \underbrace{\left(\ell_{S_{t+1}} - \ell_{S_t}\right)}_{=:B_2}
\end{aligned} \tag{175}$$

We will bound each term $B_1$ and $B_2$ separately. The first term can be easily estimated:

$$B_1 \leq 1 - \frac{\eta}{2}\left((S^*)^2 + S^*\ell_{S_\infty} + \frac{2d\lambda}{\ell_0}\right) = 1 - \eta M \tag{176}$$

where we defined

$$M := \frac{(S^*)^2 + S^*\ell_{S_\infty} + (2d\lambda)^{\frac{2}{3}}}{2} \tag{177}$$

(recall that $\ell_{S_0} = \sqrt[3]{2d\lambda}$).

In order to estimate $B_2$, we will first prove a series of useful lemmas that illustrate some properties of the sequence $\{\ell_{S_t}\}$.

**Lemma 30.** $\forall\, t \geq 0$,

$$0 < \frac{2d\lambda}{\ell_{S_\infty}} \leq \ell_{S_t}^2 - S_t \leq \frac{2d\lambda}{\ell_{S_0}}. \tag{178}$$

*Proof.* By definition of $\ell_{S_t}$,

$$P(\ell_t, S_t) = 2d\lambda - \ell_{S_t}^3 + \ell_{S_t}S_t = 0 \tag{179}$$

i.e.

$$\ell_{S_t}^2 - S_t = \frac{2d\lambda}{\ell_{S_t}} > 0. \tag{180}$$

Using the fact that $\{\ell_{S_t}\}$ is increasing and bounded ($\ell_{S_0} \leq \ell_{S_t} \leq \ell_{S_\infty}$, $\forall\, t \geq 0$), the result follows. $\qquad\square$

**Lemma 31.** $\forall\, t \geq 0$,

$$x_{t+1} - x_t \leq \eta M\left(\ell_{S_t} - x_t\right) \tag{181}$$

*with $M$ defined in (177).*

*Proof.* The proof easily follows from the recurrence relation of the (increasing) sequence $\{x_t\}$:

$$
\begin{aligned}
x_{t+1} - x_t &= \frac{\eta}{2}\left(2d\lambda - x_t^3 + x_t S_t\right) \\
&\leq (\ell_{S_t} - x_t) \cdot \underbrace{\frac{\eta}{2}\left((S^*)^2 + S^*\ell_{S_\infty} + (2d\lambda)^{\frac{2}{3}}\right)}_{=\eta M}
\end{aligned}
\tag{182}
$$

$\square$

**Lemma 32.** $\forall\, t \geq 0$,

$$
\ell_{S_{t+1}} - \ell_{S_t} \leq C_\infty \left(x_{t+1} - x_t\right)^2
\tag{183}
$$

*for some constant $C_\infty \in \mathbb{R}_{>0}$.*

*Proof.* Using again the definition of $\ell_{S_t}$, we have

$$
\begin{aligned}
\ell_{S_{t+1}}^3 - \ell_{S_t}^3 &= \ell_{S_{t+1}} S_{t+1} - \ell_{S_t} S_t \\
&= \left(\ell_{S_{t+1}} - \ell_{S_t}\right) S_{t+1} + \ell_{S_t}\left(S_{t+1} - S_t\right) \\
&= \left(\ell_{S_{t+1}} - \ell_{S_t}\right) S_{t+1} + \ell_{S_t}\left(x_{t+1} - x_t\right)^2 ;
\end{aligned}
\tag{184}
$$

on the other hand,

$$
\ell_{S_{t+1}}^3 - \ell_{S_t}^3 = \left(\ell_{S_{t+1}} - \ell_{S_t}\right)\left(\ell_{S_{t+1}}^2 + \ell_{S_{t+1}}\ell_{S_t} + \ell_{S_t}^2\right) ;
\tag{185}
$$

combining (184) and (185) and using Lemma 30, we obtain

$$
\begin{aligned}
\ell_{S_{t+1}} - \ell_{S_t} &= \frac{\ell_{S_t}\left(x_{t+1} - x_t\right)^2}{\ell_{S_{t+1}}^2 + \ell_{S_{t+1}}\ell_{S_t} + \ell_{S_t}^2 - S_{t+1}} \\
&\leq \frac{\ell_{S_\infty}\left(x_{t+1} - x_t\right)^2}{2\ell_{S_0}^2 + \frac{2d\lambda}{\ell_{S_\infty}}} = \underbrace{\frac{\ell_{S_\infty}}{2(2d\lambda)^{\frac{2}{3}} + \frac{2d\lambda}{\ell_{S_\infty}}}}_{=:C_\infty}\left(x_{t+1} - x_t\right)^2 .
\end{aligned}
\tag{186}
$$

$\square$

Collecting the results from Lemmas 31 and 32, we have

$$
B_2 \leq C_\infty (x_{t+1} - x_t)^2 \leq \eta^2 M C_\infty \left(\ell_{S_t} - x_t\right)^2
\tag{187}
$$

In conclusion, for $\eta < \frac{1}{M}$ we have

$$
\begin{aligned}
\ell_{S_{t+1}} - x_{t+1} &\leq (\ell_{S_t} - x_t)(1 - \eta M) + \eta^2 M^2 C_\infty (\ell_{S_t} - x_t)^2 \\
&= (\ell_{S_t} - x_t)\left(1 - \eta M + \eta^2 \tilde{M}\right) \\
&\leq \sqrt[3]{2d\lambda}\left(1 - \eta M + \eta^2 \tilde{M}\right)^t
\end{aligned}
\tag{188}
$$

where $\tilde{M} := 2\ell_{S_\infty} M^2 C_\infty > 0$ (we used the fact that $\{\ell_{S_t}\}$ and $\{x_t\}$ are bounded); we stress that the quantity $1 - \eta M - \eta^2 \tilde{M}$ is smaller than 1 (thus being indeed a convergence rate) for $\eta$ small enough.

# E  EXPERIMENTS

## E.1  SOLUTION OF THE FA ODE SYSTEM

Figures 1, 3, 4, 5, plotting solutions (and products of solutions) of the FA system

$$\dot{\theta}_L = (\lambda - \theta_L \dots \theta_1)\theta_{L-1}\dots\theta_1 \tag{189}$$

$$\dot{\theta}_{L-1} = d_{L-1}(\lambda - \theta_L \dots \theta_1)\theta_{L-2}\dots\theta_1 \tag{190}$$

$$\vdots \tag{191}$$

$$\dot{\theta}_2 = d_2(\lambda - \theta_L \dots \theta_1)\theta_1 \tag{192}$$

$$\dot{\theta}_1 = d_1(\lambda - \theta_L \dots \theta_1) \tag{193}$$

in the continuous setting ($t \in \mathbb{R}_{\geq 0}$) are obtained using the standard ODE solver `ode45` on Matlab (version R2020b).

## E.2  LINEAR AUTOENCODERS

We set up two linear autoencoders

$$\hat{\boldsymbol{y}} = \boldsymbol{W}_2\boldsymbol{W}_1\boldsymbol{x} \qquad \text{(2-layer NN)} \tag{194}$$

$$\hat{\boldsymbol{y}} = \boldsymbol{W}_3\boldsymbol{W}_2\boldsymbol{W}_1\boldsymbol{x} \qquad \text{(3-layer NN} \tag{195}$$

where $\boldsymbol{W}_\ell \in \mathbb{R}^{d\times d}$, $d = 20$ and the data are synthetically generated as

$$\boldsymbol{x}_i = \boldsymbol{A}\boldsymbol{z}_i + \boldsymbol{\epsilon}_i, \qquad \boldsymbol{z}_i \sim \mathcal{N}(\boldsymbol{0}, \boldsymbol{I}_h), \ \boldsymbol{\epsilon}_i \sim 10^{-3}\mathcal{N}(\boldsymbol{0}, \boldsymbol{I}_d) \tag{196}$$

$h = 5$ and $\boldsymbol{A} \in \mathbb{R}^{d\times h}$ is a fixed matrix that we sampled with entries $\boldsymbol{A}_{ij} \sim \mathcal{U}([0, 1])$.

For the experiments showed in Figure 2, we set the step size $\eta = 0.01$ and we initialized the weight matrices as $\boldsymbol{W}_1^{(0)}, \boldsymbol{W}_2^{(0)}, \boldsymbol{W}_3^{(0)}$ such that $\boldsymbol{W}_\ell^{(0)} \sim 10^{-5}\mathcal{U}([0, 1])$.

The FA matrices $\boldsymbol{M}_\ell$ are generated as $\boldsymbol{M}_{\ell;ij} \sim \mathcal{U}([0, 1])$. We repeat the FA training for 15 times, sampling a different set of FA matrices each time (but same initial conditions $\boldsymbol{W}_\ell^{(0)}$ and same matrix $\boldsymbol{A}$): in Figure 2 we then report the average of the trace norm and the reconstruction error with error bars calculated as mean $\pm$ 2·standard_deviation (the factor 2 is to make the error bars more visible in the final plot). We complement these experiments with the corresponding GD training.

