# OpenReview forum: "Convergence Analysis and Implicit Regularization of Feedback Alignment for Deep Linear Networks"
_ICLR.cc/2022/Conference — ICLR 2022 Submitted_

### Official Review · Reviewer_Q9E4 · 2021-10-26

**Correctness:** 4
**Technical Novelty And Significance:** 3
**Empirical Novelty And Significance:** 2
**Recommendation:** 5
**Confidence:** 4

**Main Review:**

This paper is clearly written. I checked most of the proofs and they seem correct. Moreover, some results, e.g. “backward learning”, are novel and interesting.

My major concern is the significance of the results since all of them are under spectral initialization. Specifically:

1. I believe not all linear regression problems admit spectral initialization. It is only possible when $\Sigma_{xx}$ and $\Sigma_{xy}$ are co-diagonalizable, i.e. the decomposition the author has should have both $\Lambda_{xx}$ and $\Lambda_{xy}$ diagonal. The introduction of spectral initialization in Section 2 is misleading to me because it is written in a way that suggests one can do spectral initialization for any linear regression problem. The author should clarify this.

2. Spectral initialization alone is too restrictive to understand the training algorithm. In practice, such a spectral initialization scheme is rarely used to train a neural network (Note: an entirely different form of spectral initialization is used in matrix completion problem), and instead one uses mostly random initialization. In my opinion, spectral initialization allows us to somehow theoretically understand phenomena that happen during practical training. In that sense, analysis for spectral initialization alone is insufficient because we don’t know whether the phenomena observed in this paper, monotonicity of the product, backward learning, etc. also happen under non-spectral initialization.
The author discussed the relaxation of the spectral assumption at the end of page 4. but I find it unconvincing without empirical evidence.

3. Following the point above, I think the author may want to add more experiments under non-spectral initialization to illustrate how should we understand these phenomena studied for spectral initialization. For example, monotonicity of the product can only happen for scalar dynamics, is there any related notion in the non-spectral case? Can backward learning happen under non-spectral initialization? Remember that the forward learning in Saxes’14 is observed empirically when weights are randomly initialized with small scale (non-spectral).

Aside from the concern on significance, I think the scalar dynamics in this paper can be understood better using visual tools. For the scalar dynamics (6), one can plot: 1) all the equilibrium points lie on the hyperbola $\theta_1\theta_2=\lambda$; 2) the trajectory of the dynamics is constrained on the parabola $\theta_2=\frac{1}{2d}\theta_1^2+K$, where $K$ depends on initialization. This makes some statements in the paper much easier to understand. For example, the flow function in (13) has three real roots if the parabola, determined by $d,K$, intersects with the hyperbola at three points. I believe the non-monotonicity of the product can also be illustrated.

Reference:
Andrew M Saxe, James L Mcclelland, and Surya Ganguli. Exact solutions to the nonlinear dynamics of learning in deep linear neural network. In International Conference on Learning Representations, 2014.

**Summary Of The Paper:**

This paper studies the Feedback Alignment (FA) algorithm on deep linear networks. In continuous time, FA algorithm is an alternative to Gradient Flow (GF) In particular, to compute the time derivative of the weights, FA algorithm starts with the gradient of the weights at each layer, and replace the part that depends on the weights of succeeding layers by a fixed matrix.

The FA algorithm on deep linear networks under spectral initialization is studied regarding both convergence and implicit bias, and the analysis are provided for both continuous-time and discrete-time FA algorithm.

**Summary Of The Review:**

My initial rating is "weak reject" because:
Strength:
1. clear writing
2. detailed analysis
3. interesting observations

Weakness:
1. Significance, need connections to initialization schemes used in practice
2. The results are hard to interpret in general non-spectral settings
3. little numerical example
4. presentation can be improved by visual tools

---

> ### Author Response · Authors · 2021-11-19
> **Response to Reviewer Q9E4**
>
> We would like to thank Reviewer Q9E4 for their review.
>
> We appreciated that they found our paper “clearly written”, that they checked most of the proofs (and they seem correct). And that they found some results “novel and interesting”.
>
> In the following, we address their questions and concerns:
>
> **Q:** *I believe not all linear regression problems admit spectral initialization. The introduction of spectral initialization in Section 2 is misleading to me because it is written in a way that suggests one can do spectral initialization for any linear regression problem.*
>
> **A:** You are absolutely right. We have not been very clear on that point in the previous version of the submission. We provide an updated discussion in the revision (Section 2, page 3).
>
> **Q:** *This paper can be understood better using visual tools. For the scalar dynamics (6), one can plot: 1) all the equilibrium points lie on the hyperbola $\theta_1\theta_2=\lambda$ ; 2) the trajectory of the dynamics is constrained on the parabola $\theta_2 = \frac{1}{2d}\theta_1 + K$, where $K$ depends on initialization. This makes some statements in the paper much easier to understand. For example, the flow function in (13) has three real roots if the parabola, determined by $d$, $K$, intersects with the hyperbola at three points. I believe the non-monotonicity of the product can also be illustrated.*
>
> **A:** Thanks a lot for this great suggestion. We believe that good illustrations provide significant help to understand complex dynamics such as the FA one. In our revision of the submission, we added a figure (Fig 1a, page 4) illustrating the dynamics of FA in the phase space $(\theta_1,\theta_2)$: in that space, we can see that some non trivial interaction dynamics arise for small values of $K$ (since the parabola indeed intersect the solution hyperbola in three points) and the non-monotonicity of $|\theta_1\theta_2- \lambda|$ for $K\neq 0$ is evident; non-monotonicity is also illustrated in Fig 1c (see also discussion in Section 2.1, page 4). However, even if this picture provides very good insights on the dynamics it does not tell the whole story:
> * It is not clear from Fig 1a that in the case $K= 0$ the quantity $|\theta_1\theta_2- \lambda|$ is monotonous.
> * Fig 1a does not give insight on the speed of convergence of the dynamics and thus cannot give intuition on the (anti)-implicit regularization phenomena.
>
> **Q:** *I think the author may want to add more experiments under non-spectral initialization to illustrate how we should understand these phenomena studied for spectral initialization. For example, monotonicity of the product can only happen for scalar dynamics, is there any related notion in the non-spectral case? Can backward learning happen under non-spectral initialization?*
>
> **A:**  Our experiments in Fig 2 are performed without spectral initialization. Thus, we can see that we can still see a notion of incremental learning happening in the situation where matrices are not initialized to be spectrally aligned. Also, it seems to indicate that backward learning does not occur under the non-spectral initialization we considered (i.e. small random Gaussian initialization).

---

### Official Review · Reviewer_6Die · 2021-11-01

**Correctness:** 4
**Technical Novelty And Significance:** 4
**Empirical Novelty And Significance:** Not applicable
**Recommendation:** 8
**Confidence:** 5

**Main Review:**

This paper is a nice theoretical development, extending the theory of linear
neural networks to the case of DFA. The authors first study the shallow case
before extending their arguments to the deep case. Throughout, they are clear
about the assumptions they make (choice of initialisation, commutativity of the
weight matrices, etc.)

I found the results concerning the implicit bias interesting - it is surprising
that FA would exhibit this inverse learning order, and I am not aware of an
implicit bias that would depend on the initial conditions in such a
qualitatively different way. This observation should be interesting beyond FA.

The numerical experiments nicely illustrate the results, and show very good
agreement with the theory, even if some of the assumptions (e.g. on
initialisation) are relaxed in the experiments.

The paper is well-structured and clearly written.

Further comments / questions:

- In the analysis of the discrete update equations, what exactly is the
  motivation for the additional term $W_1^{t+1/2}$ in Eq. 25? Is this system of
  equations simply more convenient for the analysis? If so, how does it connect
  to the algorithm?

- In your conclusion, you state that you anticipate that FA will "remove" some
  stationary points from the landscape of non-linear networks. This connection
  didn't become clear to me, could you elaborate?

Some cosmetic comments:

- Could you clarify over which distribution the the average is taken in Prop. 1?
  Is it w.r.t. the finite dataset or over the distribution $\mathcal{D}$?
- I would increase the fontsize of some of the plots to match the captions, in
  particular Fig 1.


**Summary Of The Paper:**

The authors study feedback alignment (FA, Lillicrap '14), an alternative
algorithm to the standard backpropagation algorithm to train neural
networks. The key idea is to replace the weight matrices of the network with
fixed, random "feedback matrices" when computing the weight updates while
training the network. A precise understanding how such a learning rule can lead
to learning remains an open question, and has recently attracted some interest
(see related works)

Here, the authors study the dynamics of learning with FA in (deep) linear neural
networks. The dynamics of linear neural networks trained with backpropagation
has been studied extensively, thus providing an ideal test bed for this
study. The authors derive a set of continuous-time equations governing the
dynamics of FA for linear networks, discussing in detail their assumptions
etc. They use these equations to discover an interesting implicit bias of FA:
for certain initialisations, features of the data are learnt in the *inverse*
order of importance, quantified by the singular value associated with each mode
of the data. For other initialisations, features are learnt in the order of
importance, as expected. The sensitivity of this bias to the initialisation
doesn't appear in backprop. Finally, the authors also study linear auto-encoders
trained with FA, and find that not FA does not only recover (a rotation of) the
eigenvectors, but that it also speeds up training significantly. They also
study the discrete-step dynamics of FA.

**Summary Of The Review:**

Feedback alignment has attracted some interest in recent years after it was
shown that it can train a range of neural networks, and some groups have started
investigating the mechanism that allow the training to proceed with random
matrices. This work is thus timely and should be of interest to the ICLR
community. It provides several interesting contributions regarding (1) the
mechanism behind feedback alignment algorithms and (2) the implict bias of
different learning algorithms, through an analysis of linear neural networks, a
workhorse in neural network theory. I therefore recommend acceptance at the
conference.

PS: I am not sure I fully understood the questionnaire below regarding the
difference between "technical novelty..." and "empirical novelty". Since this a
theory paper, I only voted on the technical novelty.

---

> ### Author Response · Authors · 2021-11-19
> **Response to Reviewer 6Die**
>
> We thank the referee for their feedback and for their support.
> We will address their comments below:
>
> **Q:** *What is the motivation for the additional term $W^{(t+1/2)}$ in Eq. 25?*
>
> **A:** Regarding the term $W_1^{(t+1/2)}$ in equation (25), we were inspired by the fact that the behavior of the second layer depends on the choices that we impose on the first layer, due to the initialization scheme that we propose (formula (7)), therefore we made the update at time $t+1$ of the second weight matrix to depend not only on the previous iterate $W_j^{(t)}$, but also on the newly computed $W_1^{(t+1)}$.
> Such a discretized algorithm can be also regarded as a modification of the standard Euler method, where the discretization scheme is changed in order to control the $\eta^2$ perturbation term.
> Indeed, with this strategy the convergence analysis greatly simplifies and it can be easily extended to the multiple-layer case. The proof of convergence for the forward Euler method in the 2-layer case is quite challenging; we are convinced it is possible to prove the same result also for deep networks following the same guidelines, i.e. proving that the iterative map is contracting and that specific quantities governing the update rule remain within a bounded region at any iteration. We did not pursue this direction as it was too technical, but numerical experiments agree with our intuition.
>
>
> **Q:** *You anticipate that FA will "remove" some stationary points from the landscape of non-linear networks.*
>
> **A:** In the linear setting, the equation of motions (equation (3)) with full rank feedback matrix $M$ clearly show that the only stationary points of the system are the set of weights $W_1$, $W_2$ such that $W_2W_1 = \Sigma_{xy}$, i.e. the global minima of the loss. This is in contrast with GD where in general there may exist many stationary points of the system which are not minima (for instance when all the weights are zero). In this sense, FA “removes” suboptimal critical points. Based on this observation, we are conjecturing that also in the non-linear setting, the presence of a fixed full-rank matrix that drives the dynamics, will have the benefit of guiding the weight trajectories away from spurious stationary points (some insights can be also found in [Refinetti et al., ‘20]).
>
>
> **Q:** *Over which distribution the average is taken in Prop. 1?*
>
> **A:** The average in Prop. 1 is taken with respect to any data distribution $\mathcal D$: $\mathcal D$ can be regarded as the true distribution of the data (in this case, we aim at optimizing the population risk) or as the empirical distribution in the case of the finite dataset $\mathcal D = \frac{1}{N} \sum_i \delta_{(x_i,y_i)}$ (thus, optimizing the empirical risk).

---

### Official Review · Reviewer_xi3S · 2021-11-02

**Correctness:** 4
**Technical Novelty And Significance:** 2
**Empirical Novelty And Significance:** Not applicable
**Recommendation:** 5
**Confidence:** 3

**Main Review:**

It provides convergence results on deep linear neural networks for feedback alignment for both discrete and continuous cases.
The authors show that the feedback alignment algorithm sequentially fits the solutions of a reduced-rank regression problem, and an initialization scheme is proposed to address the problem of anti-regularization.
They also show that the convergence results hold for networks that are not necessarily over-parametrized.

Feedback alignment method is named after one of its outstanding characteristics that the forward weights get aligned with the random backward weights. However, this paper does not seem to have even mentioned any attempts on such explorations.
The current results focus primarily on the linear networks, which leaves a gap between the theory and the empirical successes on networks with non-linear activation functions.
The definition of the loss function assumes a strong condition on infinite sample size.
One of the key steps of reducing the original system of ODEs into a scalar system relies on the assumption that the initialization of the weight matrix for the second layer is deterministic given the one for the first layer. However, people often initialize each layer randomly and independently before training, and the condition seems to be more artificial on deep networks.

The overall structure is clear and well-structured.

**Summary Of The Paper:**

The authors prove the linear convergence rate of the feedback alignment algorithm on fully connected linear networks. They also identify the implicit anti-regularization phenomenon for certain initialization of the algorithm and further propose initialization schemes that provide a form of implicit regularization and facilitate the learning process.

**Summary Of The Review:**

The paper is nicely written, and the phenomenon of anti-regularization is interesting. However, the convergence results seem to assume strong conditions that only works for specific cases.

---

> ### Author Response · Authors · 2021-11-19
> **Response to Reviewer xi3S**
>
> We thank the Reviewer for their observations and comments. However, we respectfully disagree on some of their comments:
>
> **Q:** *This paper does not seem to have even mentioned any attempts on such (alignment) explorations.*
>
> **A:** We do address this topic in Section 2 right after Theorem 2. It appears that the diagonal weight  $\theta_i$ of the first layer learn $(2r_i\lambda_i)^{1/3})$ that depends on the values of the feedback matrices $d_i$ and in some sense get aligned with this matrix ($\theta_i$ get small if $d_i$ is small and conversely). We added some details in the revision (Section 2.1, page 3).
>
>
> **Q:** *The current results focus primarily on the linear networks, which leaves a gap between the theory and the empirical successes on networks with non-linear activation functions.*
>
> **A:** There is a plethora of insightful works on linear networks, published at ICLR ([Arora, et al. 2018], [Hardt, Ma, 2017])  and other major conferences (e.g. [Eftekhari, 2020], [Arora, et al. 2019], [Nar, Sastry, 2018], [Bartlett, et al. 2018], [Gunasekar, et al. 2017], [Kawaguchi, 2016]). The study of linear networks is a crucial starting point for a systematic analysis of the FA algorithm. We are convinced that our contribution is useful at validating several prior numerical pieces of evidence (listed in the introduction), at least at a "first approximation" level.
>
>
> **Q:** *The definition of the loss function assumes a strong condition on infinite sample size.*
>
> **A:** Our result holds for any distribution. After question (3), we explicitly point out that the results hold for finite sample size.
>
>
> **Q:** *People often initialize each layer randomly and independently before training, and the condition seems to be more artificial on deep networks.*
>
> **A:** We address this point in the paper and have experiments by initializing randomly and independently the layers.
> In particular, in Theorem 2 we initialize the second layer in a deterministic fashion according to the first layer, however *we prove in Theorem 3 that convergence of the FA dynamics is guaranteed even if the initialization of $W_2(0)$ is independent from $W_1(0)$* (still in the diagonal setting). An analogous result also holds for deep linear networks, but due to the space limit and readability, we did not include it in the paper (a proof is now available in Appendix B).

---

### Official Review · Reviewer_eeMk · 2021-11-03

**Correctness:** 4
**Technical Novelty And Significance:** 2
**Empirical Novelty And Significance:** 2
**Recommendation:** 5
**Confidence:** 3

**Main Review:**

This paper focuses on the theoretical analysis of FA. The authors find a setting (linear networks with diagonalizable matrices) to simplify the original problem to one-dimensional dynamics, then analyze the convergence rigorously. An interesting implicit anti-regularization is also observed under certain initialization. The numerical experiment, despite the simple setting, shows some advantages of FA compared to GD.

I am not familiar with the related work about FA, but my main concern is that the one-dimensional dynamics may oversimplify the original problem; and for $L > 2$, the convergence is proved only for a special initialization.
* Compared to GD, even for linear networks, the gap between (diagonalizable) one-dimensional scalar dynamics and high-dimensional matrix dynamics is non-trivial. May we get similar results as GD (given the FA dynamics seems to be simpler than GD)?
* I tend to think that the implicit anti-regularization phenomenon is more like an interesting math problem under the special initialization. Do we observe the behavior in numerical experiments with commonly-used initialization?

As discussed in the last section, the ultimate goal of the theoretical analysis of FA is to prove its potential advantage over GD. I am afraid the current one-dimensional analysis is still far from filling this gap.

**Summary Of The Paper:**

This paper theoretically analyzes the Feedback Alignment (FA, Eqn 2) algorithm in the optimization of deep linear networks.
* Under the assumption that the data matrices, FA matrices and initial weight matrices can be diagonalized simultaneously, the optimization can be divided into several one-dimensional problems, and the paper proves the convergence of both continuous and discrete dynamics.
* After the diagonalization, each eigenvalue corresponds to a one-dimensional dynamics. The authors construct two types of initialization to show that the convergence for the larger eigenvalues can be either faster (implicit regularization) or slower (implicit anti-regularization) than the smaller eigenvalues.
* Numerical comparison between FA and GD with 2- and 3-layer linear networks and random initialization. FA converges faster than GD and shares similar implicit regularization behavior.

**Summary Of The Review:**

This paper proves some theoretical optimization results about FA under some special settings, but I think the one-dimensional setting may oversimplify the original problem, thus the novelty may not be significant enough.

---

> ### Author Response · Authors · 2021-11-19
> **Response to Reviewer eeMk**
>
> We thank the Reviewer for their valuable feedback. A reply to their concerns can be found below:
>
> **Q:** *The convergence is proved only for a special initialization*
>
> **A:** Convergence for deep networks holds for any initialization -- the analysis is lengthy (as in the 2-layer case) and due to the space limit and readability, we did not include it in the paper, but we now added the proof in the appendix (Appendix B).
>
>
> **Q:** *The gap between (diagonalizable) one-dimensional scalar dynamics and high-dimensional matrix dynamics is non-trivial. May we get similar results as GD (given the FA dynamics seems to be simpler than GD)?*
>
> **A:** Implicit regularization of GD is also only done in the linear case (see [Gidel, ‘19], [Gissin, ‘19] and [Arora, 19]).
> Regarding diagonal dynamic vs matrix dynamics, we refer to [Saxe, ‘18], where the authors argue that a diagonal initialization mimics the dynamics one gets with a full matrix initialization if initialized with small weights (which is standard practice).
>
> * Gidel, Bach, Lacoste-Julien, “Implicit Regularization of Discrete Gradient Dynamics in Deep Linear Neural Networks”, NeurIPS 2019.
> * Gissin, Shalev-Shwartz, Daniely, “The Implicit Bias of Depth: How Incremental Learning Drives Generalization”,  ICLR 2020.
> * Arora, Sanjeev, et al. "Implicit regularization in deep matrix factorization." Advances in Neural Information Processing Systems 32 (2019): 7413-7424.
> * Saxe, McClelland, Ganguli, "A mathematical theory of semantic development in deep neural networks", arXiv:1810.10531, 2018.
>
> **Q:** *The implicit anti-regularization phenomenon is more like an interesting math problem under the special initialization. Do we observe the behavior in numerical experiments with commonly-used initialization?*
>
> **A:** As pointed out by Reviewer 6Die, we do believe that the anti-implicit regularization phenomenon is novel and it can be relevant to other settings and algorithms beyond FA.
> We would like to stress that such a phenomenon is not simply a fun math calculation and it helps to better understand the whole dynamics of the algorithm. In future work, we intend to further analyze its implications (and possible applications?).
> We don’t observe anti-implicit regularization in experiments with common initialization because --as we argue in the paper-- we luckily avoid this phenomenon with high probability.
>
>
> **Q:** *The ultimate goal of the theoretical analysis of FA is to prove its potential advantage over GD.*
>
> **A:**  We agree that a comparison with GD is fundamental for a complete study of FA, however, this is not the ultimate goal of this paper.
> We propose for the first time a rigorous theoretical analysis of a method that has been broadly studied in practice. In this regard, even negative theoretical results are in the interest of the community since no convergence results were previously known.
> We believe that a sound study should first gain a solid understanding of an algorithm’s behavior, before proving that it is more advantageous with respect to other options.

---

### Decision · Program_Chairs · 2022-01-20

**Decision:**

Reject

**Comment:**

This paper provides a theoretical analysis for the Feedback Alignment (FA) algorithm, an alternative to backpropagation for training deep linear neural networks. The main drawback of the analysis is that it assumes that the initial weight matrices are diagonal, which makes the dynamics of the algorithm reduce to K independent one-dimensional dynamic. Most of the reviewers feel that this assumption is too strong. Note that in many existing papers on the implicit bias/regularization of gradient descent (GD) for optimizing deep linear networks, they do not assume the initial weight matrices are diagonal. The authors provide some additional proof in appendix B during the rebuttal to try to relax the assumption on the diagonal initialization, but it is not critical clear if the same results still hold. While this paper studies a very important problem, I suggest the authors take into account the reviewers’ comments and improve the presentation/results. In addition, a comparison with the implicit bias of GD would help better position this work, as one of the reviewers suggested.